# G2D2: Gradient-Guided Discrete Diffusion for Inverse Problem Solving

**Naoki Murata**[1], **Chieh-Hsin Lai**[1], **Yuhta Takida**[1], **Toshimitsu Uesaka**[1], **Bac Nguyen**[1],
**Stefano Ermon**[2], **Yuki Mitsufuji**[1,3]
[1]**Sony AI**, [2]**Stanford University**, [3]**Sony Group Corporation**
`naoki.murata@sony.com`

Reviewed on OpenReview: `https://openreview.net/forum?id=fj23qnVifX`

## Abstract

Recent literature has effectively leveraged diffusion models trained on continuous variables as priors for solving inverse problems. Notably, discrete diffusion models with discrete latent codes have shown strong performance, particularly in modalities suited for discrete compressed representations, such as image and motion generation. However, their discrete and non-differentiable nature has limited their application to inverse problems formulated in continuous spaces. This paper presents a novel method for addressing linear inverse problems by leveraging generative models based on discrete diffusion as priors. We overcome these limitations by approximating the true posterior distribution with a variational distribution constructed from categorical distributions and continuous relaxation techniques. Furthermore, we employ a star-shaped noise process to mitigate the drawbacks of traditional discrete diffusion models with absorbing states, demonstrating that our method performs comparably to continuous diffusion techniques with a lower GPU memory consumption. Our code is available at `https://github.com/sony/g2d2`.

## 1 Introduction

Diffusion models have gained significant attention as deep generative models, achieving remarkable success in image (Sohl-Dickstein et al., 2015; Ho et al., 2020; Song et al., 2021b; Dhariwal & Nichol, 2021; Esser et al., 2024), audio (Liu et al., 2023; Chen et al., 2024a), and video generation (Ho et al., 2022a;b). These models operate by iteratively corrupting data and then learning to reverse the corruption process, ultimately generating high-quality samples from noise. In parallel with continuous diffusion models, discrete diffusion models have emerged as a compelling alternative. These models have gained traction by demonstrating notable results not only in image (Gu et al., 2022), audio (Yang et al., 2023), and text generation (Austin et al., 2021; Lou et al., 2023a) but also in more specialized areas such as motion data (Lou et al., 2023b; Pinyoanuntapong et al., 2024), protein synthesis (Gruver et al., 2024), and graph generation (Vignac et al., 2023).

Building on these advancements, researchers have made significant progress in expanding the application of diffusion models. They have explored using diffusion models, trained either directly in the pixel space or on latent representations derived from variational autoencoders (VAEs), to address inverse problems (Kawar et al., 2022; Chung et al., 2023b; Wang et al., 2023) and carry out various conditional generation tasks (Yu et al., 2023; Bansal et al., 2024; He et al., 2024) without the need for additional training. These efforts aim to use the powerful generative capabilities of diffusion models to tackle intricate problems and generate conditional outputs, all while preserving the models' original trained parameters.

The research on applying diffusion models to inverse problems and conditional generation has been primarily restricted to diffusion models trained in continuous spaces, and methods using pre-trained discrete diffusion models as priors remain limited (Gruver et al., 2024; Chen et al., 2024b; Li et al., 2024). One of

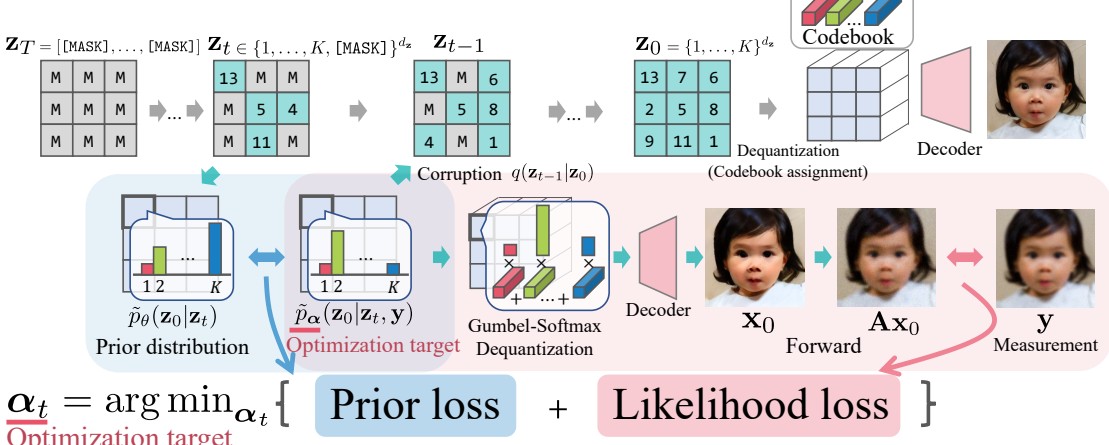

Figure 1: **Illustration of G2D2**. At each time step $t$, variational categorical distribution $\tilde{p}_{\alpha}$ is optimized with respect to the sum of prior loss and likelihood loss, followed by sampling $\mathbf{z}_{t-1}$. Both terms are continuously differentiable, enabling continuous optimization.

the main reasons is that the inherent nature of the generation process in discrete diffusion models involves non-differentiable operations, posing a challenge for their application to inverse problems formulated in continuous spaces. Therefore, controlling discrete diffusion models often necessitates an additional trained network (Gruver et al., 2024; Nisonoff et al., 2024; Klarner et al., 2024; Vignac et al., 2023). Additionally, training-free methods have been confined to relatively low-dimensional data (Chen et al., 2024b) or to specific tasks such as image inpainting (Gu et al., 2022). Recent work by Singhal et al. (2025) proposes Feynman–Kac steering, a general inference-time framework for steering diffusion models with reward functions. While this approach does not require differentiating through reward functions and thus can be applied to discrete diffusion models, the reward functions they employ, such as human preference scores, impose weaker constraints on the generation process compared to typical inverse problems like deblurring, potentially significantly compromising consistency with measurements.

Despite these limitations in applying discrete diffusion models to inverse problems, their potential advantages in representing complex data distributions and generating high-fidelity samples motivate their exploration as priors. When examining existing approaches for continuous diffusion models, conventional approaches in continuous settings typically leverage gradient-based adjustments of the generation trajectory. These methods aim to refine intermediate latents by computing gradients of a likelihood loss function, ensuring they align well with the measurement equation or guidance target. This has been demonstrated in Chung et al. (2023b) and Yu et al. (2023). However, directly extending this gradient-based control to discrete diffusion models is challenging due to their inherently non-differentiable operations.

To address the fundamental challenge of non-differentiability in discrete diffusion models, we propose Gradient-Guided Discrete Diffusion (**G2D2**), an inverse problem solving method that uses a discrete diffusion model as a prior. Our focus is on solving inverse problems using a generative model based on a discrete diffusion model, specifically designed for discrete latent variables such as those found in vector-quantized VAE (VQ-VAE) models. G2D2 bridges the gap between continuous and discrete domains by using a continuous relaxation technique to optimize the parameters of a variational distribution.

In addition to the differentiability issue, discrete diffusion models present another challenge when used for inverse problems. These models often adopt "mask-absorbing" noise processes as their data corruption process due to generation quality. However, this mask-absorbing approach has a significant drawback in inverse-problem solving. For example, in discrete diffusion models for images, while a substantial portion of the image structure is determined in the initial stages of generation (i.e., when only a few tokens are determined), the mask-absorbing type does not allow transitions from an unmasked state to either a masked state or other unmasked states. Our experiments demonstrate that this restriction imposes a significant limitation on performance in solving inverse problems.

To overcome this structural limitation of mask-absorbing processes, we incorporate the star-shaped noise process previously proposed in the context of continuous diffusion models (Okhotin et al., 2024; Zhang et al., 2024). This process removes the dependency between consecutive sampling steps, expanding the space that can be explored during generation. This process was originally proposed to enhance the performance of diffusion models (Okhotin et al., 2024), but was later introduced as a decoupled noise annealing process in the context of inverse problems using continuous diffusion models, demonstrating its effectiveness for continuous diffusion models (Zhang et al., 2024). In this study, we not only demonstrate that this process can be effectively applied to discrete diffusion models, but also discover that it uniquely addresses potential issues inherent in mask-absorbing-type discrete diffusion processes, specifically the inability to correct errors introduced in the early stages during later inference steps.

To validate our proposed approach, we conduct comprehensive experiments comparing G2D2 to current methods using standard benchmark datasets. Our results demonstrate that G2D2 achieves comparable performance to continuous counterparts (within 0.02–0.05 LPIPS points) while reducing GPU memory usage by up to 77% (4.7GiB vs 20.9GiB for PSLD). We also explore the application of a discrete prior-based motion-data-generation model to solve an inverse problem, specifically path-conditioned generation, without requiring further training. The results of our study indicate that G2D2 shows promise in tackling various inverse problems by leveraging pre-trained discrete diffusion models.

## 2 Preliminaries

### 2.1 Discrete diffusion models for image generation

We first provide a brief overview of VQ-Diffusion (Gu et al., 2022; Tang et al., 2022), an image-generation model based on discrete diffusion processes. VQ-Diffusion generates images in a two-step process. It first produces discrete latent representations $\mathbf{z}_0$ using a discrete diffusion model trained on representations obtained from a pre-trained VQ-VAE model (van den Oord et al., 2017). It then transforms these representations into the continuous image space using a decoder. Each element of $\mathbf{z}_0 \in \{1, \ldots, K\}^{d_\mathbf{z}}$ corresponds to one of the embedding vectors from the codebook, denoted as $\mathbf{B} := \{\mathbf{b}_1, \ldots, \mathbf{b}_K\}, \mathbf{b}_k \in \mathbb{R}^{d_\mathbf{b}}$. During decoding, a variable $\mathbf{Z} \in \mathbf{B}^{d_\mathbf{z}}$ is constructed through codebook assignment, where $(\mathbf{Z})_i = \mathbf{b}_{z_{0,i}}$ and $z_{0,i}$ denotes the $i$-th element of $\mathbf{z}_0$. This variable is then fed into a decoder $D : \mathbb{R}^{d_\mathbf{b} \times d_\mathbf{z}} \to \mathbb{R}^{d_{\mathbf{x}_0}}$ that maps from the discrete token embeddings to the continuous image space to obtain the final image: $\mathbf{x}_0 = D(\mathbf{Z})$.

In discrete diffusion models, a forward Markov process gradually corrupts the discrete latent representation $\mathbf{z}_0$, and a reverse process is learned to invert this process. A single step of the forward process of the Markov chain $\mathbf{z}_0 \to \cdots \to \mathbf{z}_t \to \cdots \to \mathbf{z}_T$ can be represented as,

$$q(z_{t,i}|\mathbf{z}_{t-1}) = \boldsymbol{v}^\mathsf{T}(z_{t,i})Q_t\boldsymbol{v}(z_{t-1,i}), \tag{1}$$

where $\boldsymbol{v}(z_{t,i}) \in \{0,1\}^{K+1}$ denotes a one-hot encoded vector representing the token at time step $t$, $K$ is the number of states from the VQ-VAE, and one additional state is for a special mask token [MASK]. These concepts will be formally introduced immediately below in the context of VQ-Diffusion. $Q_t \in \mathbb{R}^{(K+1)\times(K+1)}$ represents the transition matrix, which determines the probabilities of transitions between tokens. VQ-Diffusion uses a mask-absorbing-type forward process, which introduces a special masked token denoted as [MASK] in addition to the $K$ states from the VQ-VAE. The transition matrix is defined as follows, where the last column represents the transition probabilities for the [MASK] token:

$$Q_t = \begin{pmatrix} \alpha_t + \beta_t & \beta_t & \beta_t & \cdots & 0 \\ \beta_t & \alpha_t + \beta_t & \beta_t & \cdots & 0 \\ \beta_t & \beta_t & \alpha_t + \beta_t & \cdots & 0 \\ \vdots & \vdots & \vdots & \ddots & \vdots \\ \gamma_t & \gamma_t & \gamma_t & \cdots & 1 \end{pmatrix}, \tag{2}$$

where the transition probabilities are determined by three parameters: $\alpha_t$, $\beta_t$, and $\gamma_t$. In this process, once a token transitions to the masked state, it remains masked in all subsequent steps, hence the term

"absorbing." Specifically, $\alpha_t$ represents the probability of a token remaining unchanged, $\beta_t$ denotes the probability of transitioning to a different unmasked token, and $\gamma_t$ indicates the probability of the token being replaced with the [MASK] token. These parameters satisfy the constraint $\alpha_t + (K-1)\beta_t + \gamma_t = 1$ to ensure valid probability distributions. The probability $\beta_t$ between unmasked tokens is generally set to a very small value. These parameters are typically set so that $q(\mathbf{z}_T|\mathbf{z}_0)$ assigns all probability mass to the [MASK] token, and we also adopt this assumption.

During inference, the latent variable $\mathbf{z}_0$ corresponding to the clean image is obtained by executing the following reverse process:

$$p_\theta(\mathbf{z}_{t-1}|\mathbf{z}_t) = \sum_{\mathbf{z}_0} q(\mathbf{z}_{t-1}|\mathbf{z}_t, \mathbf{z}_0)\tilde{p}_\theta(\mathbf{z}_0|\mathbf{z}_t), \tag{3}$$

where $q(\mathbf{z}_{t-1}|\mathbf{z}_t, \mathbf{z}_0)$ represents the posterior distribution determined by the forward process, and $\tilde{p}_\theta$ denotes the denoising network that predicts the denoised token distribution at $t$. The output of $\tilde{p}_\theta$ is generally modeled as independent categorical distributions for each dimension in $\mathbf{z}_0$. In practical applications such as text-to-image generation, $\tilde{p}_\theta$ is trained with conditional information (e.g., text prompts in VQ-Diffusion), allowing for controlled generation. While the true data distribution $q(\mathbf{z}_0)$ has dependencies across dimensions, the denoising network $\tilde{p}_\theta(\mathbf{z}_0|\mathbf{z}_t)$ typically models each dimension independently as categorical distributions. However, the complete reverse process defined in (3) implicitly captures some of these dependencies through the iterative application of the conditional distributions $p_\theta(\mathbf{z}_{t-1}|\mathbf{z}_t)$. We distinguish between two distributions: the clean distribution $\tilde{p}_\theta(\mathbf{z}_0|\mathbf{z}_t)$ directly estimated by the denoising network at a single step (which treats dimensions independently), and the distribution $p_\theta(\mathbf{z}_0|\mathbf{z}_t)$ obtained by running the reverse diffusion process from $t$ to $t = 0$, which better approximates the true data distribution with its dimensional dependencies.

## 2.2 Linear-inverse-problem settings

Inverse problems involve estimating unknown data from measurement. The relationship between the measurement $\mathbf{y} \in \mathbb{R}^{d_\mathbf{y}}$ and unknown ground-truth data $\mathbf{x}_0 \in \mathbb{R}^{d_{\mathbf{x}_0}}$ can be represented as

$$\mathbf{y} = \mathbf{A}\mathbf{x}_0 + \boldsymbol{\eta}, \tag{4}$$

where $\mathbf{A} \in \mathbb{R}^{d_\mathbf{y} \times d_{\mathbf{x}_0}}$ is referred to as the forward linear operator, which describes the process by which the measurement $\mathbf{y}$ is obtained from data $\mathbf{x}_0$. We assume this operator is known. The term $\boldsymbol{\eta}$ represents measurement noise, which we assume follows an isotropic Gaussian distribution with a known variance $\sigma_{\boldsymbol{\eta}}^2$. Consequently, the likelihood function $q(\mathbf{y}|\mathbf{x}_0)$ can be described as $\mathcal{N}(\mathbf{y}; \mathbf{A}\mathbf{x}_0, \sigma_{\boldsymbol{\eta}}^2\mathbf{I})$.

One of the primary challenges in inverse problems is their ill-posed nature. This means that for any given measurement $\mathbf{y}$, multiple candidate solutions may exist. To address this issue and determine $\mathbf{x}_0$, a common approach is to assume a prior distribution for $\mathbf{x}_0$, such as a Laplace distribution. Diffusion models have been utilized as more powerful and expressive priors, offering enhanced capabilities in solving these inverse problems (Kawar et al., 2022; Chung et al., 2023b; Wang et al., 2023; Rout et al., 2023). These diffusion-based methods are able to produce data that not only fit the measurement data but also exhibit high likelihood under the prior model. Given a prior $q(\mathbf{x}_0)$, the objective in the inverse problem is to sample from the posterior distribution $q(\mathbf{x}_0|\mathbf{y})$, which, according to Bayes' theorem, is proportional to $q(\mathbf{y}|\mathbf{x}_0)q(\mathbf{x}_0)$.

These methods can be categorized based on how they incorporate the information from the measurement $\mathbf{y}$ into the generation trajectory of diffusion models. Methods such as denoising diffusion restoration models (DDRM) (Kawar et al., 2022) and denoising diffusion null-space models (DDNM) (Wang et al., 2023) leverage the assumption of linear operators, using singular value decomposition of the forward process to control the generative process. In contrast, methods such as diffusion posterior sampling (DPS) (Chung et al., 2023b) and posterior sampling with latent diffusion (PSLD) (Rout et al., 2023) operate by propagating the gradient of a loss term through the generative process. This loss term is designed to maximize the measurement likelihood, specifically by minimizing the term $\|\mathbf{y} - \mathbf{A}\mathbf{x}_0\|_2^2$.

However, while these methods work well with continuous diffusion models, their application to generative models that use discrete diffusion models as priors is not straightforward. This limitation stems from two

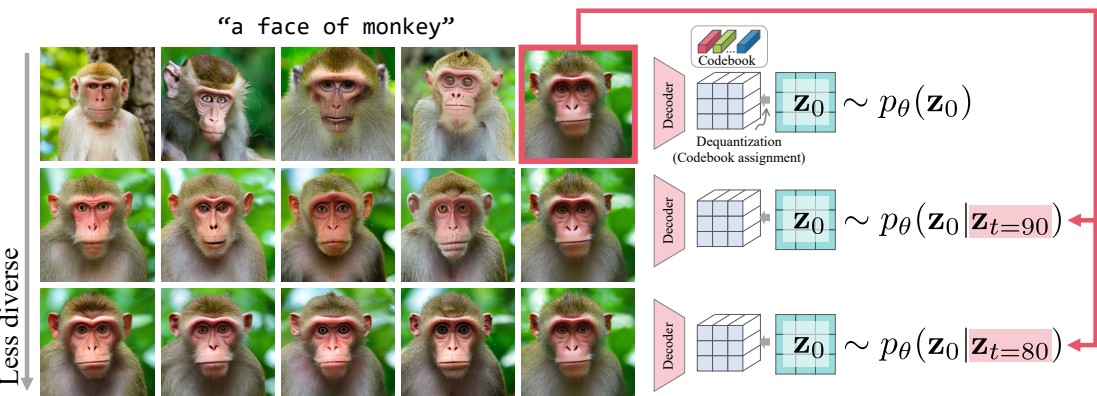

Figure 2: Empirical demonstration of mask-absorbing characteristics in discrete diffusion models. All images are generated using the text prompt "a face of monkey". Top row: Diverse samples from text-conditioned sampling ($p_\theta(\mathbf{z}_0)$). Middle row: Multiple samples conditioned on the same $\mathbf{z}_{t=90}$ ($p_\theta(\mathbf{z}_0|\mathbf{z}_{t=90})$). Bottom row: Multiple samples conditioned on the same $\mathbf{z}_{t=80}$ ($p_\theta(\mathbf{z}_0|\mathbf{z}_{t=80})$). This demonstrates that in mask-absorbing processes, image structure is largely determined in early sampling steps, with decreasing diversity as sampling progresses, making it difficult to correct inconsistencies with measurement data during inverse problem solving.

primary factors. First, the former methods (DDRM and DDNM) are specifically designed for diffusion models trained directly in the pixel domain. Second, while the latter methods (DPS and PSLD) can be extended to latent diffusion models that operate in continuous latent spaces, they encounter difficulties when handling discrete diffusion models, where the generative process involves inherently non-differentiable operations. The core challenge lies in the lack of a direct mechanism to propagate gradients of the loss function through the generative process in discrete diffusion models. In such models, after generating discrete data, a non-differentiable operation (i.e., codebook assignment) is followed by a decoding operation into continuous space, which prevents the application of conventional gradient-based guidance.

## 3 Gradient-Guided Discrete Diffusion, G2D2

In this section, we propose *Gradient-Guided Discrete Diffusion* (**G2D2**), an inverse problem solving method that leverages discrete diffusion models as priors. Our approach addresses two key challenges: the non-differentiability of discrete diffusion models and the limitations imposed by widely-used mask-absorbing noise processes. First, we introduce a star-shaped noise process to address the latter issue and define a more tractable variant of the star-shaped noise process distribution, which we call the star-decomposed distribution. We then address the former challenge of non-differentiability by bridging the gap between discrete and continuous domains through optimizing a variational distribution with continuous relaxation. These methodological components are integrated into a unified, practical inference algorithm.

### 3.1 Star-shaped noise process for enabling inherent re-masking

A key challenge in applying discrete diffusion models to inverse problems is addressing the limitations of mask-absorbing noise processes. As described in Section 2.1, discrete diffusion models commonly adopt a mask-absorbing process, where in the forward process, an unmasked token either remains the same or transitions to a mask token. While this design leads to higher performance in standard generation tasks, it poses a significant constraint in solving inverse problems. Specifically, in the reverse process (i.e., the generative process), once an unmasked token has been set at an early stage of sampling, the probability of it reverting to a masked state or changing to a different token becomes extremely low (see Figure 3b), making it difficult to correct errors made in the early stages.

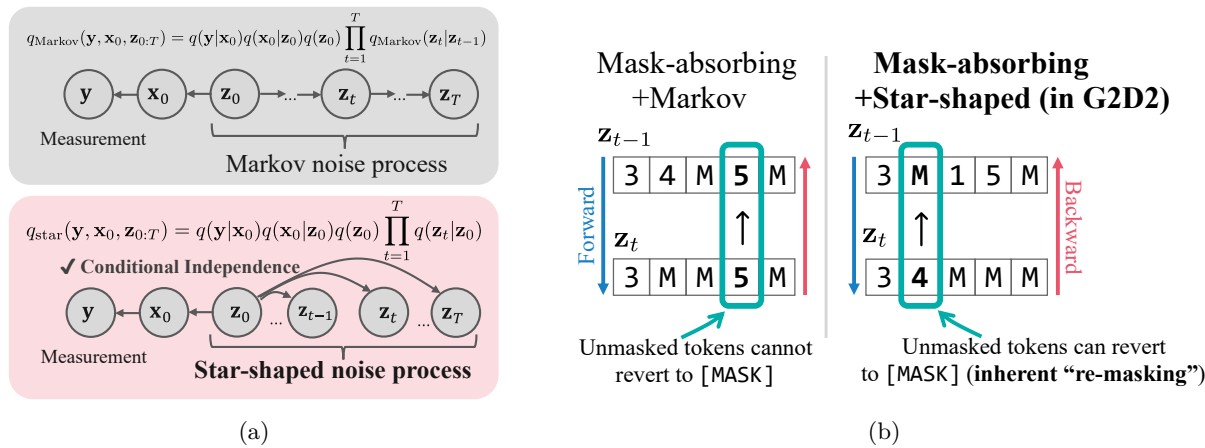

(a)               (b)

Figure 3: (a) Graphical model comparison between Markov noise process (top) and star-shaped noise process (bottom). The star-shaped model introduces conditional independence between $\mathbf{z}_t$ variables given $\mathbf{z}_0$, enabling more flexible error correction during sampling. (b) Comparison between mask-absorbing Markov process (left) and mask-absorbing star-shaped process (right). In the Markov process, unmasked tokens cannot revert to [MASK], while the star-shaped process allows tokens to naturally revert to [MASK] state (inherent "re-masking"), facilitating error correction in later sampling steps.

Figure 2 demonstrates this characteristic: when generating images using discrete diffusion models that employ mask-absorbing process, most of the image structure is determined in the initial stages, suggesting that in inverse problem scenarios, there is little opportunity to correct inconsistencies with the measurement $\mathbf{y}$ during sampling. One solution to this problem is the "re-masking" operation, which reverts unmasked tokens to masked tokens. This approach has been implemented in discrete predictor-corrector methods (Lezama et al., 2023) and predictor-corrector techniques for continuous-time discrete diffusion (Campbell et al., 2022; Zhao et al., 2024) to improve image generation quality. However, these methods often involve reverting the sampler to earlier time steps and can be computationally expensive.

Here, drawing inspiration from Okhotin et al. (2024) and Zhang et al. (2024), we adopt a *star-shaped noise process* that decouples each time step $t$ from previous steps, which addresses the aforementioned issue of early-stage errors becoming permanently "frozen." In this star-shaped noise process, the noisy variables $\mathbf{z}_1, \ldots, \mathbf{z}_T$ are conditionally independent given $\mathbf{z}_0$. Formally, we assume $q_{\text{star}}(\mathbf{z}_{1:T}|\mathbf{z}_0) = \prod_{t=1}^{T} q_{\text{star}}(\mathbf{z}_t|\mathbf{z}_0) = \prod_{t=1}^{T} \prod_{i=1}^{d_\mathbf{z}} \left[ \boldsymbol{v}^{\mathsf{T}}(z_{t,i}) \overline{Q}_t \, \boldsymbol{v}(z_{0,i}) \right]$, where the matrix $\overline{Q}_t = Q_t \cdots Q_1$ is the cumulative transition from the original Markov forward process in discrete diffusion, but the forward graphical model is star-shaped, as shown in Figure 3a.

Compared to the original Markov forward process, the star-shaped noise process maintains the same conditional marginal distribution $q_{\text{star}}(\mathbf{z}_t|\mathbf{z}_0)$, yet the locations of [MASK] tokens are uncorrelated between $\mathbf{z}_t$ corresponding to different time steps. In the Markov noise process, once a token is [MASK]ed, it remains so, and in subsequent steps, only unmasked tokens can transition to [MASK] tokens. In contrast, in the star-shaped noise process, each noisy token $\mathbf{z}_t$ is viewed as being generated directly from $\mathbf{z}_0$ rather than from $\mathbf{z}_{t-1}$, making the [MASK] token positions independent between adjacent $\mathbf{z}_{t-1}$ and $\mathbf{z}_t$.

A key characteristic of the star-shaped process is observed in the reverse step $q_{\text{star}}(\mathbf{z}_{t-1}|\mathbf{z}_t)$. Unlike the Markov noise process with mask-absorbing state, which requires that tokens already unmasked remain fixed and prohibits their reversion to a masked state, the star-shaped formulation permits tokens to transition from an unmasked state back to a masked state within a single reverse step. This inherent "re-masking" operation enables flexible error correction during sampling, as illustrated in Figure 3b, where tokens can naturally revert to the masked state without the need to roll back the sampler.

Our goal here, similar to other diffusion model-based inverse problem methods, is to sample from $q_{\text{star}}(\mathbf{z}_0|\mathbf{y})$ based on this graphical model. More specifically, we aim to develop a feasible algorithm for sampling from $q_{\text{star}}(\mathbf{z}_{0:T}|\mathbf{y})$. However, attempting to perform ancestral sampling from this distribution, such as by

proceeding from $\mathbf{z}_T \to \mathbf{z}_{T-1} \to \cdots \to \mathbf{z}_0$, results in a posterior of the form $q_{\text{star}}(\mathbf{z}_t|\mathbf{z}_{t+1:T}, \mathbf{y})$, which depends on all subsequent samples and is therefore not computationally feasible.

To address this challenge, we first introduce a proxy joint distribution, $q_{\text{star-decomp}}(\mathbf{z}_{0:T}|\mathbf{y})$, obtained by factorizing the star-shaped posterior into a product of local conditionals. Importantly, this preserves the marginal $q_{\text{star-decomp}}(\mathbf{z}_0|\mathbf{y}) = q_{\text{star}}(\mathbf{z}_0|\mathbf{y})$ as we will discuss in the next subsection, so it can serve as a *valid surrogate* for $q_{\text{star}}$ while reducing the complexity of the graph structure so that dependencies exist only between consecutive time steps. Although the full joint $q_{\text{star-decomp}}$ is still intractable, this decomposition is a necessary preparatory step that leads to the variational approximation described in Sec. 3.3. The resulting algorithm separates the complex inter-dependencies between time steps while maintaining the essential properties of the star-shaped model, making it possible to efficiently sample from $q_{\text{star}}(\mathbf{z}_0|\mathbf{y})$.

### 3.2 Defining the decomposed star-shaped distribution $q_{\text{star-decomp}}$ which is a tractable variant of $q_{\text{star}}$

We define the *star-decomposed* distribution as follows:

$$q_{\text{star-decomp}}(\mathbf{z}_{0:T}|\mathbf{y}) = q_{\text{star}}(\mathbf{z}_T|\mathbf{y}) \prod_{t=1}^{T} q_{\text{star}}(\mathbf{z}_{t-1}|\mathbf{z}_t, \mathbf{y}), \tag{5}$$

where $q_{\text{star}}(\mathbf{z}_{t-1}|\mathbf{z}_t, \mathbf{y})$ represents the conditional distribution of $q_{\text{star}}$. This decomposition of the joint distribution resembles the Markovian reverse process of diffusion models, but it has the following properties:

**1. A single step of $q_{\text{star-decomp}}$ inherently enables the "re-masking" operation.** In the star-shaped noise process, the positions of mask tokens in $\mathbf{z}_{t-1}$ and $\mathbf{z}_t$ are mutually independent and uncorrelated as shown in Figure 3b. Consequently, the conditional distribution $q_{\text{star-decomp}}(\mathbf{z}_{t-1}|\mathbf{z}_t, \mathbf{y})$ $(= q_{\text{star}}(\mathbf{z}_{t-1}|\mathbf{z}_t, \mathbf{y}))$ enables a "re-masking" operation, wherein unmasked tokens present in $\mathbf{z}_t$ can become masked tokens in $\mathbf{z}_{t-1}$. This property suggests that in mask-absorbing discrete diffusion, errors that occur in the initial stages of sampling can be corrected in subsequent steps, which provides an advantage when solving inverse problems.

**2. The marginal distribution $q_{\text{star-decomp}}(\mathbf{z}_0|\mathbf{y})$ is identical to the target distribution $q_{\text{star}}(\mathbf{z}_0|\mathbf{y})$.** The statement and proof are provided in the Appendix. This suggests that for solving inverse problems, we do not necessarily need to sample from the joint distribution of $q_{\text{star}}$, but can instead aim to sample from the more tractable distribution $q_{\text{star-decomp}}$. Ultimately, we perform approximate sampling from the posterior by approximating this $q_{\text{star-decomp}}$ using a variational distribution.

**3. The conditional joint distribution of $q_{\text{star-decomp}}$ differs from that of $q_{\text{star}}$, i.e., $q_{\text{star-decomp}}(\mathbf{z}_{0:T}|\mathbf{y}) \neq q_{\text{star}}(\mathbf{z}_{0:T}|\mathbf{y})$.** The decomposition of the joint distribution of a star-shaped noise process takes the form $q_{\text{star}}(\mathbf{z}_{0:T}|\mathbf{y}) = q_{\text{star}}(\mathbf{z}_T|\mathbf{y}) \prod_{t=1}^{T} q_{\text{star}}(\mathbf{z}_{t-1}|\mathbf{z}_{t:T}, \mathbf{y})$. However, $q_{\text{star-decomp}}$ deviates from this formulation by disregarding the dependencies on larger time steps, $\mathbf{z}_{t+1:T}$.

In subsequent sections, we introduce a variational distribution to approximate $q_{\text{star-decomp}}$, which possesses Property 1 that inherently enables the re-masking operation. As established by Property 3, the joint distribution of $q_{\text{star-decomp}}$ differs from that of the star-shaped noise process graphical model. Nevertheless, Property 2 guarantees that they share identical marginal distributions given the measurement $\mathbf{y}$, providing justification for our approach of targeting the more tractable $q_{\text{star-decomp}}$ for sampling. Furthermore, since $q_{\text{star-decomp}}$ focuses only on two adjacent variables, we can formulate a simple algorithm to approximate its distribution using a variational approach.

### 3.3 Variational approximation for feasible inference algorithm

Based on the discussion in the previous section, we aim to implement $q_{\text{star-decomp}}$, which inherently incorporates a re-masking process for efficient inverse problem solving. However, since $q_{\text{star-decomp}}$ is still not tractable, we introduce a variational distribution $p_{\boldsymbol{\alpha}}(\mathbf{z}_{0:T}|\mathbf{y}) = q_{\text{star}}(\mathbf{z}_T|\mathbf{y}) \prod_{t=1}^{T} p_{\boldsymbol{\alpha}}(\mathbf{z}_{t-1}|\mathbf{z}_t, \mathbf{y})$ to approximate $q_{\text{star-decomp}}(\mathbf{z}_{0:T}|\mathbf{y})$, with the ultimate goal of ensuring that the marginal distribution $p_{\boldsymbol{\alpha}}(\mathbf{z}_0|\mathbf{y})$

approximates the true posterior $q_{\text{star-decomp}}(\mathbf{z}_0|\mathbf{y})$. The distribution $p_{\boldsymbol{\alpha}}$ is decomposed as

$$p_{\boldsymbol{\alpha}}(\mathbf{z}_{t-1}|\mathbf{z}_t, \mathbf{y}) = \sum_{\mathbf{z}_0} q_{\text{star}}(\mathbf{z}_{t-1}|\mathbf{z}_0)\tilde{p}_{\boldsymbol{\alpha}}(\mathbf{z}_0|\mathbf{z}_t, \mathbf{y}), \tag{6}$$

where $\tilde{p}_{\boldsymbol{\alpha}}(\mathbf{z}_0|\mathbf{z}_t, \mathbf{y})$ is a categorical distribution parameterized by $\boldsymbol{\alpha}$, where for each time step $t$ and dimension $i$, $\boldsymbol{\alpha}_{t,i,\cdot}$ is a probability vector in the simplex $\Delta^{K-1}$, defined as $\tilde{p}_{\boldsymbol{\alpha}}(z_{0,i}|\mathbf{z}_t, \mathbf{y}) = \text{Cat}\left(z_{0,i}; \boldsymbol{\alpha}_{t,i,\cdot}\right)$, i.e., $\tilde{p}_{\boldsymbol{\alpha}}(z_{0,i} = k|\mathbf{z}_t, \mathbf{y}) = \alpha_{t,i,k}$. In the subsequent discussions and optimization steps, we assume that $\alpha_{t,i,\cdot}$ is always normalized to lie on the simplex. This decomposition stems from the fact that the distribution $q_{\text{star-decomp}}(\mathbf{z}_{t-1}|\mathbf{z}_t, \mathbf{y})$ $(= q_{\text{star}}(\mathbf{z}_{t-1}|\mathbf{z}_t, \mathbf{y}))$ can be expressed as $\sum_{\mathbf{z}_0} q_{\text{star}}(\mathbf{z}_{t-1}|\mathbf{z}_0)q_{\text{star}}(\mathbf{z}_0|\mathbf{z}_t, \mathbf{y})$ based on the conditional independence. Note that both $q_{\text{star}}(\mathbf{z}_{t-1}|\mathbf{z}_0)$ and $\tilde{p}_{\boldsymbol{\alpha}}(\mathbf{z}_0|\mathbf{z}_t, \mathbf{y})$ have a mean field structure with independent categorical distributions across dimensions. Consequently, $p_{\boldsymbol{\alpha}}(\mathbf{z}_{t-1}|\mathbf{z}_t, \mathbf{y})$, obtained by marginalizing over $\mathbf{z}_0$, inherits this mean field property. For notational convenience, we denote the slice of distribution parameter $\boldsymbol{\alpha}$ at time step $t$ as $\boldsymbol{\alpha}_t \in \mathbb{R}^{d_{\mathbf{z}} \times K}$.

To ensure that the marginal distribution of the variational distribution $p_{\boldsymbol{\alpha}}$ closely approximates that of $q_{\text{star-decomp}}$, the parameters $\boldsymbol{\alpha}$ of the variational distribution are obtained by optimizing an objective function derived from the following theorem:

**Theorem 3.1.** *Let $p_{\boldsymbol{\alpha}}$ be a distribution with the parameterization given by the decomposition in (6). Then, for any measurements $\mathbf{y}$, the following inequality holds for the KL divergence between the marginal distributions:*

$$D_{\text{KL}}\left(p_{\boldsymbol{\alpha}}(\mathbf{z}_0|\mathbf{y})\|q_{star\text{-}decomp}(\mathbf{z}_0|\mathbf{y})\right) \le \sum_{t=1}^{T} \mathbb{E}_{\mathbf{z}_t \sim p_{\boldsymbol{\alpha}}(\mathbf{z}_t|\mathbf{y})}\left[D_{\text{KL}}\left(\tilde{p}_{\boldsymbol{\alpha}}(\mathbf{z}_0|\mathbf{z}_t, \mathbf{y})\|q_{star}(\mathbf{z}_0|\mathbf{z}_t, \mathbf{y})\right)\right], \tag{7}$$

*where $p_{\boldsymbol{\alpha}}(\mathbf{z}_0|\mathbf{y})$ is the variational marginal distribution parameterized by $\boldsymbol{\alpha}$, $q_{star}(\mathbf{z}_0|\mathbf{y})$ is the true posterior distribution, $\tilde{p}_{\boldsymbol{\alpha}}(\mathbf{z}_0|\mathbf{z}_t, \mathbf{y})$ is the variational conditional distribution as defined in (6), $q_{star}(\mathbf{z}_0|\mathbf{z}_t, \mathbf{y})$ is the true conditional distribution, $p_{\boldsymbol{\alpha}}(\mathbf{z}_t|\mathbf{y})$ is the marginal distribution at time step $t$, and $T$ is the total number of time steps in the diffusion process.*

The proof is provided in the Appendix. Based on this inequality, we aim to minimize each term in the sum on the right-hand side. Since $\tilde{p}_{\boldsymbol{\alpha}}$ is a different categorical distribution at each $t$, we minimize $\boldsymbol{\alpha}$ for each time step, ultimately aiming to minimize the left-hand side. Each term on the right-hand side of (7) can be decomposed into a term representing deviation from the prior and a term representing likelihood with respect to the observed data:

**Lemma 3.1.** *The KL divergence between the variational distribution $\tilde{p}_{\boldsymbol{\alpha}}(\mathbf{z}_0|\mathbf{z}_t, \mathbf{y})$ and the true posterior $q_{star}(\mathbf{z}_0|\mathbf{z}_t, \mathbf{y})$ can be decomposed into two terms:*

$$D_{\text{KL}}\left(\tilde{p}_{\boldsymbol{\alpha}}(\mathbf{z}_0|\mathbf{z}_t, \mathbf{y})\|q(\mathbf{z}_0|\mathbf{z}_t, \mathbf{y})\right) = D_{\text{KL}}\left(\tilde{p}_{\boldsymbol{\alpha}}(\mathbf{z}_0|\mathbf{z}_t, \mathbf{y})\|q_{star}(\mathbf{z}_0|\mathbf{z}_t)\right) - \mathbb{E}_{\mathbf{z}_0 \sim \tilde{p}_{\boldsymbol{\alpha}}(\mathbf{z}_0|\mathbf{z}_t, \mathbf{y})}\left[\log q_{star}(\mathbf{y}|\mathbf{z}_0)\right], \tag{8}$$

This decomposition enables us to separately consider **the fit to the prior** and **the consistency with the measurement data**, and we approximate both terms in a computationally tractable form as follows. First, the KL term on the right-hand side of (8) remains intractable. However, we note that the star-shaped noise process shares the conditional distribution $q_{\text{star}}(\mathbf{z}_t|\mathbf{z}_0)$ with the original Markov noise process. Consequently, the reverse conditional distribution $q_{\text{star}}(\mathbf{z}_0|\mathbf{z}_t)$ will also be identical for both processes. Since the prior of the pre-trained discrete diffusion models is trained to approximate this distribution, we substitute this prior model $\tilde{p}_{\theta}(\mathbf{z}_0|\mathbf{z}_t)$ for $q_{\text{star}}(\mathbf{z}_0|\mathbf{z}_t)$ into the objective function of (8). This substitution transforms the term into a KL divergence between two categorical distributions, enabling the computation of gradients with respect to the parameter $\boldsymbol{\alpha}$.

The second term involves an expectation calculation over a categorical distribution, for which we use the Gumbel-Softmax re-parameterization trick (Jang et al., 2016; Maddison et al., 2016). The implementation of this trick is discussed in the subsequent section. This approach makes the term differentiable with respect

to the categorical distribution's parameter $\boldsymbol{\alpha}$, facilitating continuous optimization. The explicit form of the resultant loss function is detailed in the Appendix.

Based on Theorem 3.1 and Lemma 3.1, our proposed inverse problem solving method with discrete diffusion prior, **G2D2**, optimizes the parameter $\boldsymbol{\alpha}$ of $p_{\boldsymbol{\alpha}}$ for $t = T, \ldots, 1$ while sequentially sampling $\mathbf{z}_{0:T}$. In the optimization step, any continuous optimization method, such as Adam (Kingma, 2014) and RAdam (Liu et al., 2020), can be used. Implementation considerations are discussed in the following section. This algorithm is detailed in Algorithm 1, and G2D2 is illustrated in Figure 1.

---

**Algorithm 1** Gradient-Guided Discrete Diffusion, **G2D2**

---

**Require:** Input condition $\mathbf{y}$, pre-trained discrete diffusion model $p_{\theta}$, forget coefficient $\gamma$
1: $\mathbf{z}_T \sim q_{\text{star}}(\mathbf{z}_T)$
2: **for** $t = T, \ldots, 1$ **do**
3:     **if** $t = T$ **then**
4:         Initialize: $\boldsymbol{\alpha}_t \propto \log \tilde{p}_{\theta}(\mathbf{z}_0|\mathbf{z}_t)$
5:     **else**
6:         Initialize: $\boldsymbol{\alpha}_t \propto \exp(\gamma \log \boldsymbol{\alpha}_{t+1} + (1 - \gamma) \log \tilde{p}_{\theta}(\mathbf{z}_0|\mathbf{z}_t))$
7:     **end if**
8:     `// Continuous optimization`
9:     $\boldsymbol{\alpha}_t = \arg\min_{\boldsymbol{\alpha}_t} D_{\text{KL}}\left(\tilde{p}_{\boldsymbol{\alpha}}(\mathbf{z}_0|\mathbf{z}_t, \mathbf{y}) \| \tilde{p}_{\theta}(\mathbf{z}_0|\mathbf{z}_t)\right) - \mathbb{E}_{\mathbf{z}_0 \sim \tilde{p}_{\boldsymbol{\alpha}}(\mathbf{z}_0|\mathbf{z}_t, \mathbf{y})}\left[\log q_{\text{star}}(\mathbf{y}|\mathbf{z}_0)\right]$
10:     Sample $\mathbf{z}_{t-1} \sim p_{\boldsymbol{\alpha}}(\mathbf{z}_{t-1}|\mathbf{z}_t, \mathbf{y}) = \sum_{\mathbf{z}_0} q_{\text{star}}(\mathbf{z}_{t-1}|\mathbf{z}_0)\tilde{p}_{\boldsymbol{\alpha}}(\mathbf{z}_0|\mathbf{z}_t, \mathbf{y})$
11: **end for**
12: **return** $\mathbf{x}_0$ by decoding $\mathbf{z}_0$

---

### 3.4 Implementation considerations

**Gumbel-Softmax dequantization** We use the Gumbel-Softmax trick (Jang et al., 2016; Maddison et al., 2016) to make the computation of the second term in (8) differentiable. At time step $t$, this process begins by generating Gumbel-Softmax samples using parameters of $\tilde{p}_{\boldsymbol{\alpha}}$ as follows: $\hat{z}_{0,i,k} = \text{softmax}\left((\log \alpha_{t,i,k} + g_{i,k})/\tau\right)$, where $g_{i,k}$ are i.i.d. samples drawn from the Gumbel distribution, and $\tau$ ($> 0$) is the temperature parameter. This procedure generates a "soft" categorical sample for each dimension in $\mathbf{z}_0$, indicating the proportional selection of each codebook element. As these proportions correspond to the contribution rate of each codebook element, we construct $\mathbf{Z}_{\text{Gumbel}} \in \mathbb{R}^{d_{\mathbf{z}} \times d_b}$ as their weighted sum: $(\mathbf{Z}_{\text{Gumbel}})_i = \sum_{k=1}^K \hat{z}_{0,i,k} \mathbf{b}_k$. Finally, we pass $\mathbf{Z}_{\text{Gumbel}}$ through the decoder to obtain the image $\mathbf{x}_0 = D(\mathbf{Z}_{\text{Gumbel}})$. By substituting this image into the likelihood function $q_{\text{star}}(\mathbf{y}|\mathbf{x}_0)$, we obtain the differentiable objective with respect to the variational parameter $\boldsymbol{\alpha}_t$, enabling continuous optimization. For linear inverse problems, the objective function will include the term $\|\mathbf{y} - \mathbf{A}\mathbf{x}_0(\boldsymbol{\alpha}_t)\|_2^2$, excluding the constant term derived from measurement noise.

**Optimization initialization strategy** At time step $t$, we are required to optimize the variational parameter $\boldsymbol{\alpha}_t$. To expedite this process, we can leverage the optimized values from the previous time step as the initialization for the optimization process, effectively reducing the number of required optimization steps. To achieve this, we introduce a forgetting coefficient $\gamma$ (where $0 \leq \gamma \leq 1$) and initialize $\boldsymbol{\alpha}_t$ through a weighted sum of the previous optimized variables and the prior model's output in the logarithm domain, given by $\boldsymbol{\alpha}_t \propto \exp(\gamma \log \boldsymbol{\alpha}_{t+1} + (1 - \gamma) \log \tilde{p}_{\theta}(\mathbf{z}_0|\mathbf{z}_t))$. The effectiveness of this strategy is discussed in Appendix E.6.

### 3.5 Application of G2D2 to masked generative models

As discussed in (Zheng et al., 2024), mask-absorbing discrete diffusion models and masked generative models, such as MaskGIT (Chang et al., 2022), share a similar framework. Apart from temporal conditioning, these models are nearly identical and are trained to approximate $q_{\text{star}}(\mathbf{z}_0|\mathbf{z}_t)$. Therefore, G2D2 can be straightforwardly applied to masked generative models. We empirically show this by providing an example of solving inverse problems using a masked generative model as a prior for motion data in the experimental section.

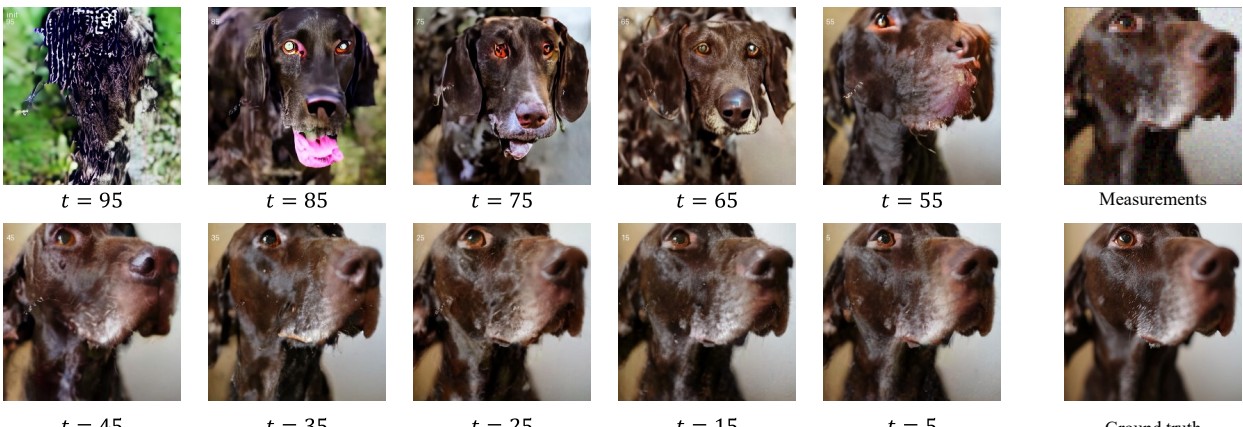

| $t = 95$ | $t = 85$ | $t = 75$ | $t = 65$ | $t = 55$ | Measurements |

| $t = 45$ | $t = 35$ | $t = 25$ | $t = 15$ | $t = 5$ | Ground truth |

Figure 4: Images sampled from the prior model $\tilde{p}_\theta(\mathbf{z}_0|\mathbf{z}_t)$ using intermediate $\mathbf{z}_t$ during the process of G2D2 in image inverse problem solving. The progression demonstrates how initial structural errors are gradually corrected as the sampling proceeds in G2D2.

## 4 Experiments

### 4.1 Experimental setup

We evaluate G2D2 on inverse problems in the image processing and compare it with other diffusion model-based inverse-problem-solving methods. We also demonstrate gradient-based guidance on a discrete-latent variable-based motion-domain generative model without additional training, showing the applicability of G2D2 to other domains.

**Image inverse problems and evaluation metrics**  We conduct experiments on two tasks: (1) super-resolution (SR) and (2) Gaussian deblurring. For the SR task, the linear forward operator downscales the image by a factor of 4 using a bicubic resizer. For the Gaussian deblurring task, we set the kernel size to $61 \times 61$ with a Gaussian kernel standard deviation of 3.0. The measurements are obtained by applying the forward operator to the ground truth images normalized to the range $[-1, 1]$, followed by the addition of Gaussian noise with a standard deviation of 0.05. As metrics, we use the learned perceptual image patch similarity (LPIPS) (Zhang et al., 2018) score to measure perceptual proximity to the original image, and the peak signal-to-noise ratio (PSNR) to measure the closeness of the signal.

**Datasets**  Following previous studies, we use the ImageNet (Deng et al., 2009) and Flickr-Faces-HQ (FFHQ) (Karras et al., 2019) datasets. The images are $256 \times 256$. For comparison, we use a subset of 1000 images from each validation set.

**Baselines**  We compare DPS (Chung et al., 2023b), DDRM (Kawar et al., 2022), which use diffusion models trained in the pixel domain, and PSLD (Rout et al., 2023) and ReSample (Song et al., 2024), which use diffusion models trained in the latent space acquired from VAE (latent diffusion models) as baselines with G2D2.

**Implementation details**  Regarding G2D2, for both the ImageNet and FFHQ experiments, we use a pretrained VQ-Diffusion model [1] that is trained on the ITHQ dataset (Tang et al., 2022). In all experiments, we optimize the parameters $\boldsymbol{\alpha}_t$ of the variational categorical distribution within the G2D2 algorithm's optimization step using the RAdam optimizer (Liu et al., 2020). To balance the prior and likelihood terms in the objective function, we introduce hyperparameters. For the image inverse problem experiments, we used text prompts for the VQ-Diffusion model: "`a photo of [Class Name]`" for ImageNet and "`a high-quality`

---

[1]`https://huggingface.co/microsoft/vq-diffusion-ithq`

`headshot of a person`" for FFHQ. All experiments are performed on one RTX 3090 (24 GiB), but G2D2 itself never exceeded 4.7 GiB of VRAM and required 194 s per ImageNet image (Table 5 in the Appendix). Hence the full pipeline can be executed on widely available 8 GiB cards.

Details of the experiments and comparison methods are provided in the Appendix.

## 4.2 Image inverse problem solving on ImageNet and FFHQ

Figure 5 shows the qualitative results of image inverse problem solving, and Tables 1 and 2 list the quantitative results. As shown in Tables 1 and 2, G2D2 performs comparably to the methods using diffusion models trained in continuous domains. Notably, G2D2 achieves this performance while consuming significantly lower GPU memory (4.7GiB compared to 10.7GiB for DPS and 20.9GiB for PSLD, as detailed in Table 5 in the Appendix) and maintaining competitive computational speed among gradient-based methods. Note that the pre-trained models used for each method are different, which particularly contributes to the superiority of pixel-domain methods on FFHQ. With DDRM, it is assumed that the amount of measurement noise is known, and it requires the singular value decomposition of the linear operator. We also show images in the intermediate steps of the G2D2 algorithm in Figure 4.

Table 1: Quantitative evaluation on ImageNet 256×256. Performance comparison of different methods on various linear tasks in image domain. Values show the mean over 1000 images.

| Prior Type | Method | SR (×4) | | Gaussian Deblurring | |
|---|---|---|---|---|---|
| | | LPIPS(↓) | PSNR(↑) | LPIPS(↓) | PSNR(↑) |
| Pixel-domain | DPS (Chung et al., 2023b) | 0.362 | 22.67 | 0.432 | 19.49 |
| | DDRM (Kawar et al., 2022) | 0.351 | 24.27 | 0.250 | 27.61 |
| LDM | PSLD (Rout et al., 2023) | 0.331 | 24.02 | 0.359 | 23.95 |
| | ReSample (Song et al., 2024) | 0.373 | 23.31 | 0.425 | 23.07 |
| Discrete | G2D2 (proposed) | 0.340 | 23.82 | 0.367 | 23.37 |
| | G2D2 w/ Markov noise process | 0.442 | 22.01 | 0.424 | 22.36 |

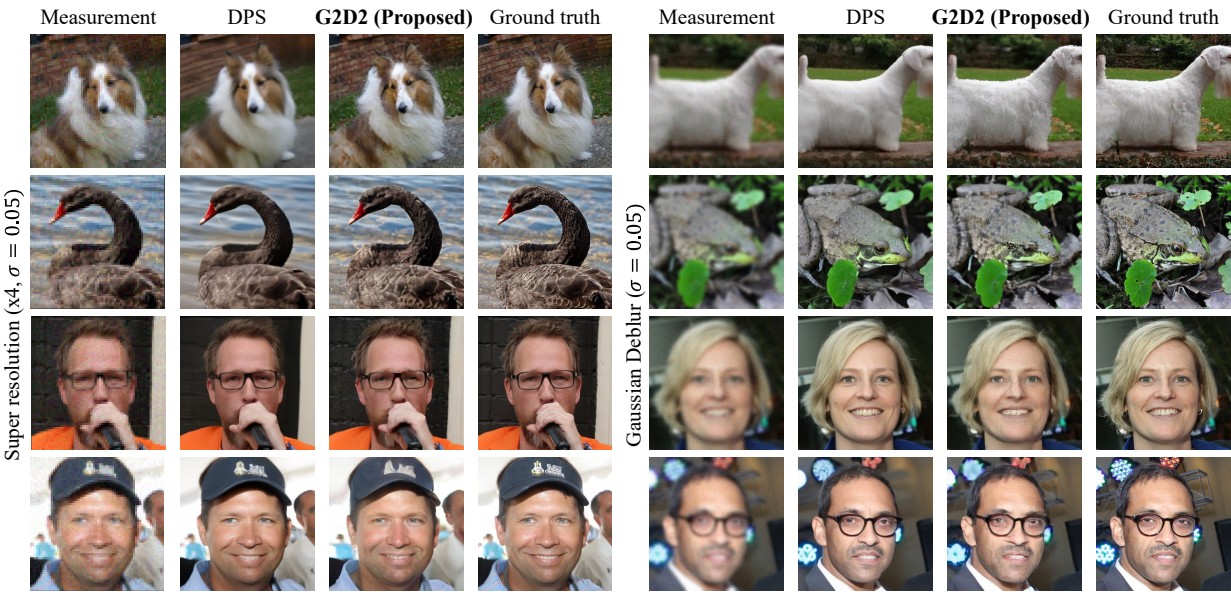

Figure 5: Qualitative results of G2D2 and DPS.

Table 2: Quantitative evaluation on FFHQ 256×256. Performance comparison of different methods on various linear tasks in image domain. Values show the mean over 1000 images.

| Prior Type | Method | SR (×4) | | Gaussian Deblurring | |
|---|---|---|---|---|---|
| | | LPIPS(↓) | PSNR(↑) | LPIPS(↓) | PSNR(↑) |
| Pixel-domain | DPS (Chung et al., 2023b) | 0.238 | 26.07 | 0.234 | 25.47 |
| | DDRM (Kawar et al., 2022) | 0.252 | 28.09 | 0.209 | 30.89 |
| LDM | PSLD (Rout et al., 2023) | 0.282 | 27.12 | 0.307 | 26.96 |
| | ReSample (Song et al., 2024) | 0.508 | 23.07 | 0.336 | 25.91 |
| Discrete | G2D2 (proposed) | 0.265 | 27.29 | 0.280 | 26.91 |
| | G2D2 w/ Markov noise process | 0.369 | 25.15 | 0.340 | 25.96 |

### 4.3 Ablation study on Graphical Models

It is possible to derive a similar algorithm to G2D2 that uses a Markov noise process as the graphical model. However, as discussed at the beginning of Section 3, this graphical model does not allow for the "re-masking" operation, which means it cannot correct errors that occur early in the sampling process. We refer to this variant as **G2D2 w/ Markov noise process**, and its performance is presented in Tables 1 and 2 on ImageNet and FFHQ, respectively. Additional qualitative results are provided in the Appendix E.3. The results indicate that the introduction of the star-shaped noise process significantly improves performance, making G2D2 comparable to methods based on continuous diffusion.

### 4.4 Motion inverse problem solving

As discussed in Section 3.5, our method can also be applied to masked generative models. We conduct experiments to manipulate the Generative Masked Motion Model (MMM) (Pinyoanuntapong et al., 2024) using gradient guidance for a **path-following task** where generation is conditioned on hip joint position information. Since joint positions can be calculated from motion data, this fits within the inverse problems framework. While path following has been achieved with continuous latent space models (Song et al., 2023b; Uchida et al., 2024), we are the first to accomplish this using a discrete latent variable motion model without additional training. The Appendix E.13 provides additional samples and experimental details.

We compare G2D2 with OmniControl (Xie et al., 2024) and Guided Motion Diffusion (GMD) (Karunratanakul et al., 2023) on controllable motion generation. Following OmniControl's setup, we evaluate using the HumanML3D test set with sparse conditioning (5 frames out of 196) on metrics including: FID, R-Precision, Diversity, Foot Skating ratio, Trajectory error (50cm), Location error (50cm), and Average error.

Table 3 presents the results. While OmniControl and GMD are specifically fine-tuned for this task, G2D2 requires no additional training yet achieves good FID and foot skating scores. However, OmniControl and GMD demonstrate better trajectory and location errors, indicating room for improvement. Note that G2D2 could potentially be combined with fine-tuned approaches (as OmniControl does) and enhanced with techniques like FreeDoM's time-traveling method (Yu et al., 2023) to further improve performance.

| Method | FID (↓) | R-prec. (↑) | Diversity (9.503→) | Foot skating (↓) | Traj. Err (50cm, ↓) | Loc. err. (50cm, ↓) | Avg. err. (↓) |
|---|---|---|---|---|---|---|---|
| G2D2 | 0.248 | 0.770 | 9.381 | 0.048 | 0.272 | 0.116 | 0.230 |
| OmniControl (Xie et al., 2024) | 0.278 | 0.705 | 9.582 | 0.058 | 0.053 | 0.015 | 0.043 |
| GMD (Karunratanakul et al., 2023) | 0.523 | 0.599 | N/A | 0.086 | 0.176 | 0.049 | 0.139 |

Table 3: Comparison of methods for controllable motion generation.

## 5 Conclusion

We proposed G2D2 for solving inverse problems using discrete diffusion models as priors. We demonstrated that G2D2 effectively addresses the limitations of discrete diffusion in inverse problem-solving by using a

continuous relaxation technique and star-shaped noise process. Specifically, G2D2 approximates the posterior in inverse problems by optimizing the parameters of a variational distribution, composed of parameterized categorical distributions, at each time step of the diffusion process. Our experiments show that G2D2 performs comparably to its continuous counterparts, opening up possibilities for training-free applications of discrete diffusion models across a wide range of tasks.

**Limitations and future work**    While G2D2 already matches the image quality of continuous counterparts, its key advantage is an order-of-magnitude reduction in GPU memory footprint, enabling inference on consumer-grade 8 GiB cards. Future work will focus on further accelerating sampling speed. We expect these gaps will narrow through efficiency optimizations and stronger prior models. Future work will also explore more challenging settings, such as nonlinear inverse problems, and extend G2D2 to additional modalities, including audio and video.

**Acknowledgments**    We are grateful to our colleague Satoshi Hayakawa for providing valuable feedback prior to submission, and we also thank the area chair and the reviewers for their constructive comments.

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

# A    Ethics statement

Our G2D2 method, which uses discrete diffusion models as priors for solving inverse problems, carries potential risks similar to those of previously proposed techniques in this field. We acknowledge that these methods, including ours, may inadvertently perpetuate biases present in training data or be misused for generating misleading or harmful content. We are committed to addressing these ethical concerns and promoting responsible use of our technology. We urge users of our method to exercise caution and consider the ethical implications of its applications.

# B    Reproducibility statement

We provide a detailed description of the experimental reproduction in the Appendix, and our code is publicly available at `https://github.com/sony/g2d2`.

# C    Related work

In this section, we review the relevant prior works.

## C.1    Leveraging Diffusion Models as Prior Models for Inverse Problems

**Pixel-Domain Diffusion Models for Inverse Problems**    Several methods have been proposed that utilize pixel-domain diffusion models for solving inverse problems. DDRM and DDNM (Kawar et al., 2022; Wang et al., 2023) assume linear operators and known noise levels, leveraging the singular value decomposition (SVD) of these operators. ΠGDM (Song et al., 2023a) can handle certain classes of non-linear operators, such as low dynamic range, where a pseudo-inverse operator can be defined. Notably, ΠGDM does not require SVD or gradient computations for such a case.

DPS (Chung et al., 2023b) broadens the applicability to cases where operator gradients can be computed, enabling it to handle both linear and non-linear operators like phase retrieval and non-linear blur. Other notable methods in this category include RePaint (Lugmayr et al., 2022) and RED-Diff (Mardani et al., 2024).

**Latent Diffusion Models for Inverse Problems**    Recent work has also explored the use of latent diffusion models for inverse problems. PSLD (Rout et al., 2023) extends the ideas of DPS to latent diffusion models, demonstrating provable sample recovery for linear inverse problems. ReSample (Song et al., 2024) achieves data consistency by solving an optimization problem at each step during sampling.

Of particular relevance to our work is DAPS (Zhang et al., 2024), which, like our approach, adopts a graphical model during sampling that differs from the one used during training of the prior model. This approach, known as the noise decoupling scheme, offers new possibilities for adapting diffusion models to various inverse problems.

**Application of Inverse Problem Solving in Various Domains**    Solving inverse problems using diffusion models has enabled various real-world applications. In the image domain, diffusion models have been extensively studied and applied to tasks such as image deblurring, super-resolution, and inpainting (Lugmayr et al., 2022; Chung et al., 2023a; Zhu et al., 2023). In the audio domain, methods such as those proposed by Song et al. (2021a), Chung et al. (2023c), and Bian et al. (2024) have been developed to address tasks like dereverberation and audio restoration. Similarly, in the medical imaging domain, approaches like those introduced by Song et al. (2021a), Chung et al. (2023c), and Bian et al. (2024) have been used to improve image reconstruction and enhance diagnostic accuracy. These advancements demonstrate the versatility and effectiveness of diffusion models across different domains.

## C.2 Conditional Generation Using Discrete Diffusion Models as Priors

While our work focuses on inverse problems, it is important to consider related approaches in conditional generation tasks using discrete diffusion models as priors. These methods, primarily developed in the context of graph generation and protein design, introduce new conditioning to pre-trained models rather than directly addressing inverse problems.

The predominant strategy in this field involves training additional guidance networks. For instance, in protein sequence generation, LaMBO-2 (Gruver et al., 2024) and Cemri et al. (2024) learn networks that evaluate how intermediate features of samples during generation achieve the desired objectives. Similarly, CGD (Klarner et al., 2024) learns a guidance model for corrupted data. Other examples requiring additional training include Nisonoff et al. (2024) and DiGress (Vignac et al., 2023) for graph generation.

In contrast, Chen et al. (2024b) proposes a training-free approach to guide discrete diffusion models for generating Electronic Health Record data. This method employs Langevin dynamics sampling to minimize a given loss function by adjusting the parameters of the final layer of the prior model's transformer output. However, this approach faces scalability issues with models having large discrete latent spaces, such as VQ-Diffusion, as it requires evaluating all possible discrete states to compute the loss function.

Another training-free method for guiding generative models with discrete latents is proposed by Li et al. (2024). This approach avoids gradient computation of the loss function, instead evaluating the loss on multiple generated samples and conducting sampling based on these values. However, like Chen et al. (2024b), this method is expected to be inefficient for models with relatively large discrete latent spaces.

# D Proofs

In this section, we provide detailed proofs for the main theoretical results presented in the paper.

**Theorem 3.1.** *Let $p_{\boldsymbol{\alpha}}$ be a distribution with the parameterization given by the decomposition in (6). Then, for any measurements $\mathbf{y}$, the following inequality holds for the KL divergence between the marginal distributions:*

$$D_{\mathrm{KL}}\left(p_{\boldsymbol{\alpha}}(\mathbf{z}_0|\mathbf{y})\|q_{star\text{-}decomp}(\mathbf{z}_0|\mathbf{y})\right) \leq \sum_{t=1}^{T} \mathbb{E}_{\mathbf{z}_t \sim p_{\boldsymbol{\alpha}}(\mathbf{z}_t|\mathbf{y})}\left[D_{\mathrm{KL}}\left(\tilde{p}_{\boldsymbol{\alpha}}(\mathbf{z}_0|\mathbf{z}_t,\mathbf{y})\|q_{star}(\mathbf{z}_0|\mathbf{z}_t,\mathbf{y})\right)\right], \qquad (7)$$

*where $p_{\boldsymbol{\alpha}}(\mathbf{z}_0|\mathbf{y})$ is the variational marginal distribution parameterized by $\boldsymbol{\alpha}$, $q_{star}(\mathbf{z}_0|\mathbf{y})$ is the true posterior distribution, $\tilde{p}_{\boldsymbol{\alpha}}(\mathbf{z}_0|\mathbf{z}_t,\mathbf{y})$ is the variational conditional distribution as defined in (6), $q_{star}(\mathbf{z}_0|\mathbf{z}_t,\mathbf{y})$ is the true conditional distribution, $p_{\boldsymbol{\alpha}}(\mathbf{z}_t|\mathbf{y})$ is the marginal distribution at time step t, and T is the total number of time steps in the diffusion process.*

*Proof.* To prove the inequality in Theorem 3.1, we start by noting that the KL divergence between the marginal distributions $p_{\boldsymbol{\alpha}}(\mathbf{z}_0|\mathbf{y})$ and $q_{\text{star-decomp}}(\mathbf{z}_0|\mathbf{y})$ can be bounded by the KL divergence between the joint distributions $p_{\boldsymbol{\alpha}}(\mathbf{z}_{0:T}|\mathbf{y})$ and $q_{\text{star-decomp}}(\mathbf{z}_{0:T}|\mathbf{y})$:

$$D_{\mathrm{KL}}\left(p_{\boldsymbol{\alpha}}(\mathbf{z}_0|\mathbf{y})\|q_{\text{star-decomp}}(\mathbf{z}_0|\mathbf{y})\right) \leq D_{\mathrm{KL}}\left(p_{\boldsymbol{\alpha}}(\mathbf{z}_{0:T}|\mathbf{y})\|q_{\text{star-decomp}}(\mathbf{z}_{0:T}|\mathbf{y})\right). \qquad (9)$$

This inequality holds because marginalization cannot increase the KL divergence between distributions.

Next, we decompose the joint KL divergence using the chain rule and the definitions of the distributions:

$$D_{\mathrm{KL}}\left(p_{\boldsymbol{\alpha}}(\mathbf{z}_{0:T}|\mathbf{y})\|q_{\text{star-decomp}}(\mathbf{z}_{0:T}|\mathbf{y})\right) = \mathbb{E}_{p_{\boldsymbol{\alpha}}(\mathbf{z}_{0:T}|\mathbf{y})}\left[\log\frac{p_{\boldsymbol{\alpha}}(\mathbf{z}_{0:T}|\mathbf{y})}{q_{\text{star-decomp}}(\mathbf{z}_{0:T}|\mathbf{y})}\right]$$

$$= \mathbb{E}_{p_{\boldsymbol{\alpha}}(\mathbf{z}_{0:T}|\mathbf{y})}\left[\log\frac{p_{\boldsymbol{\alpha}}(\mathbf{z}_T|\mathbf{y})\prod_{t=1}^{T}p_{\boldsymbol{\alpha}}(\mathbf{z}_{t-1}|\mathbf{z}_t,\mathbf{y})}{q_{\text{star}}(\mathbf{z}_T|\mathbf{y})\prod_{t=1}^{T}q_{\text{star}}(\mathbf{z}_{t-1}|\mathbf{z}_t,\mathbf{y})}\right]$$

$$= \mathbb{E}_{p_{\boldsymbol{\alpha}}(\mathbf{z}_{0:T}|\mathbf{y})}\left[\log\frac{p_{\boldsymbol{\alpha}}(\mathbf{z}_T|\mathbf{y})}{q_{\text{star}}(\mathbf{z}_T|\mathbf{y})} + \sum_{t=1}^{T}\log\frac{p_{\boldsymbol{\alpha}}(\mathbf{z}_{t-1}|\mathbf{z}_t,\mathbf{y})}{q_{\text{star}}(\mathbf{z}_{t-1}|\mathbf{z}_t,\mathbf{y})}\right]$$

$$= D_{\mathrm{KL}}\left(p_{\boldsymbol{\alpha}}(\mathbf{z}_T|\mathbf{y})\|q_{\text{star}}(\mathbf{z}_T|\mathbf{y})\right) + \sum_{t=1}^{T}\mathbb{E}_{p_{\boldsymbol{\alpha}}(\mathbf{z}_{0:T}|\mathbf{y})}\left[\log\frac{p_{\boldsymbol{\alpha}}(\mathbf{z}_{t-1}|\mathbf{z}_t,\mathbf{y})}{q_{\text{star}}(\mathbf{z}_{t-1}|\mathbf{z}_t,\mathbf{y})}\right].$$

$$\tag{10}$$

In the context of mask-absorbing state diffusion, the distribution $p_{\boldsymbol{\alpha}}(\mathbf{z}_T|\mathbf{y})$ is the same as $q_{\text{star}}(\mathbf{z}_T|\mathbf{y})$ because $\mathbf{z}_T$ is fully determined by the diffusion process and is independent of $\boldsymbol{\alpha}$. Therefore, the first term is zero:

$$D_{\mathrm{KL}}\left(p_{\boldsymbol{\alpha}}(\mathbf{z}_T|\mathbf{y})\|q_{\text{star}}(\mathbf{z}_T|\mathbf{y})\right) = 0. \tag{11}$$

This simplifies (10) to:

$$D_{\mathrm{KL}}\left(p_{\boldsymbol{\alpha}}(\mathbf{z}_{0:T}|\mathbf{y})\|q_{\text{star-decomp}}(\mathbf{z}_{0:T}|\mathbf{y})\right) = \sum_{t=1}^{T}\mathbb{E}_{p_{\boldsymbol{\alpha}}(\mathbf{z}_{0:T}|\mathbf{y})}\left[\log\frac{p_{\boldsymbol{\alpha}}(\mathbf{z}_{t-1}|\mathbf{z}_t,\mathbf{y})}{q_{\text{star}}(\mathbf{z}_{t-1}|\mathbf{z}_t,\mathbf{y})}\right]. \tag{12}$$

We can further simplify the expectation over $\mathbf{z}_{0:T}$ by focusing on $\mathbf{z}_t$ and $\mathbf{z}_{t-1}$:

$$D_{\mathrm{KL}}\left(p_{\boldsymbol{\alpha}}(\mathbf{z}_{0:T}|\mathbf{y})\|q_{\text{star-decomp}}(\mathbf{z}_{0:T}|\mathbf{y})\right) = \sum_{t=1}^{T}\mathbb{E}_{\mathbf{z}_t\sim p_{\boldsymbol{\alpha}}(\mathbf{z}_t|\mathbf{y})}\left[D_{\mathrm{KL}}\left(p_{\boldsymbol{\alpha}}(\mathbf{z}_{t-1}|\mathbf{z}_t,\mathbf{y})\|q_{\text{star}}(\mathbf{z}_{t-1}|\mathbf{z}_t,\mathbf{y})\right)\right]. \tag{13}$$

Now, for each term in the sum, we apply the chain rule for KL divergence to relate $\mathbf{z}_{t-1}$ and $\mathbf{z}_0$:

$$D_{\mathrm{KL}}\left(p_{\boldsymbol{\alpha}}(\mathbf{z}_{t-1}|\mathbf{z}_t,\mathbf{y})\|q_{\text{star}}(\mathbf{z}_{t-1}|\mathbf{z}_t,\mathbf{y})\right)$$
$$+ \mathbb{E}_{\mathbf{z}_{t-1}\sim p_{\boldsymbol{\alpha}}(\mathbf{z}_{t-1}|\mathbf{z}_t,\mathbf{y})}\left[D_{\mathrm{KL}}\left(\tilde{p}_{\boldsymbol{\alpha}}(\mathbf{z}_0|\mathbf{z}_{t-1},\mathbf{z}_t,\mathbf{y})\|q_{\text{star}}(\mathbf{z}_0|\mathbf{z}_{t-1},\mathbf{z}_t,\mathbf{y})\right)\right]$$
$$= D_{\mathrm{KL}}\left(\tilde{p}_{\boldsymbol{\alpha}}(\mathbf{z}_0|\mathbf{z}_t,\mathbf{y})\|q_{\text{star}}(\mathbf{z}_0|\mathbf{z}_t,\mathbf{y})\right)$$
$$+ \mathbb{E}_{\mathbf{z}_0\sim\tilde{p}_{\boldsymbol{\alpha}}(\mathbf{z}_0|\mathbf{z}_t,\mathbf{y})}\left[D_{\mathrm{KL}}\left(p_{\boldsymbol{\alpha}}(\mathbf{z}_{t-1}|\mathbf{z}_0,\mathbf{z}_t,\mathbf{y})\|q_{\text{star}}(\mathbf{z}_{t-1}|\mathbf{z}_0,\mathbf{z}_t,\mathbf{y})\right)\right]. \tag{14}$$

In this equation, the left-hand side represents the KL divergence between $p_{\boldsymbol{\alpha}}$ and $q_{\text{star}}$ at time $t-1$ conditioned on $\mathbf{z}_t$, plus the expected KL divergence between their respective conditional distributions of $\mathbf{z}_0$. The right-hand side represents the KL divergence between $\tilde{p}_{\boldsymbol{\alpha}}$ and $q_{\text{star}}$ directly conditioned on $\mathbf{z}_t$, plus an expected KL divergence over $\mathbf{z}_0$.

The crucial observation here is that the last term on the right-hand side is zero. This is because $p_{\boldsymbol{\alpha}}$ and $q_{\text{star}}$ share the same conditional posterior when conditioned on $\mathbf{z}_0$ and $\mathbf{z}_t$, i.e.,

$$p_{\boldsymbol{\alpha}}(\mathbf{z}_{t-1}|\mathbf{z}_0,\mathbf{z}_t,\mathbf{y}) = q_{\text{star}}(\mathbf{z}_{t-1}|\mathbf{z}_0)$$
$$= q_{\text{star}}(\mathbf{z}_{t-1}|\mathbf{z}_0,\mathbf{z}_t,\mathbf{y}). \tag{15}$$

Therefore, the KL divergence between these conditional distributions is zero:

$$D_{\mathrm{KL}}\left(p_{\boldsymbol{\alpha}}(\mathbf{z}_{t-1}|\mathbf{z}_0,\mathbf{z}_t,\mathbf{y})\|q_{\mathrm{star}}(\mathbf{z}_{t-1}|\mathbf{z}_0,\mathbf{z}_t,\mathbf{y})\right)=0. \tag{16}$$

Substituting back into (14), we obtain:

$$\begin{aligned}
&D_{\mathrm{KL}}\left(p_{\boldsymbol{\alpha}}(\mathbf{z}_{t-1}|\mathbf{z}_t,\mathbf{y})\|q_{\mathrm{star}}(\mathbf{z}_{t-1}|\mathbf{z}_t,\mathbf{y})\right)\\
&=D_{\mathrm{KL}}\left(\tilde{p}_{\boldsymbol{\alpha}}(\mathbf{z}_0|\mathbf{z}_t,\mathbf{y})\|q_{\mathrm{star}}(\mathbf{z}_0|\mathbf{z}_t,\mathbf{y})\right)\\
&\quad-\mathbb{E}_{\mathbf{z}_{t-1}\sim p_{\boldsymbol{\alpha}}(\mathbf{z}_{t-1}|\mathbf{z}_t,\mathbf{y})}\left[D_{\mathrm{KL}}\left(\tilde{p}_{\boldsymbol{\alpha}}(\mathbf{z}_0|\mathbf{z}_{t-1},\mathbf{z}_t,\mathbf{y})\|q_{\mathrm{star}}(\mathbf{z}_0|\mathbf{z}_{t-1},\mathbf{z}_t,\mathbf{y})\right)\right].
\end{aligned} \tag{17}$$

Since the KL divergence is always non-negative, the expected KL divergence on the right-hand side is non-negative, which implies:

$$D_{\mathrm{KL}}\left(p_{\boldsymbol{\alpha}}(\mathbf{z}_{t-1}|\mathbf{z}_t,\mathbf{y})\|q_{\mathrm{star}}(\mathbf{z}_{t-1}|\mathbf{z}_t,\mathbf{y})\right)\le D_{\mathrm{KL}}\left(\tilde{p}_{\boldsymbol{\alpha}}(\mathbf{z}_0|\mathbf{z}_t,\mathbf{y})\|q_{\mathrm{star}}(\mathbf{z}_0|\mathbf{z}_t,\mathbf{y})\right). \tag{18}$$

Substituting (18) back into (13), we obtain an upper bound on the joint KL divergence:

$$D_{\mathrm{KL}}\left(p_{\boldsymbol{\alpha}}(\mathbf{z}_{0:T}|\mathbf{y})\|q_{\mathrm{star\text{-}decomp}}(\mathbf{z}_{0:T}|\mathbf{y})\right)\le\sum_{t=1}^{T}\mathbb{E}_{\mathbf{z}_t\sim p_{\boldsymbol{\alpha}}(\mathbf{z}_t|\mathbf{y})}\left[D_{\mathrm{KL}}\left(\tilde{p}_{\boldsymbol{\alpha}}(\mathbf{z}_0|\mathbf{z}_t,\mathbf{y})\|q_{\mathrm{star}}(\mathbf{z}_0|\mathbf{z}_t,\mathbf{y})\right)\right]. \tag{19}$$

Combining (9) and (19), we conclude:

$$D_{\mathrm{KL}}\left(p_{\boldsymbol{\alpha}}(\mathbf{z}_0|\mathbf{y})\|q_{\mathrm{star\text{-}decomp}}(\mathbf{z}_0|\mathbf{y})\right)\le\sum_{t=1}^{T}\mathbb{E}_{\mathbf{z}_t\sim p_{\boldsymbol{\alpha}}(\mathbf{z}_t|\mathbf{y})}\left[D_{\mathrm{KL}}\left(\tilde{p}_{\boldsymbol{\alpha}}(\mathbf{z}_0|\mathbf{z}_t,\mathbf{y})\|q_{\mathrm{star}}(\mathbf{z}_0|\mathbf{z}_t,\mathbf{y})\right)\right]. \tag{20}$$

This establishes the inequality stated in the theorem. $\qquad\square$

**Lemma 3.1.** *The KL divergence between the variational distribution $\tilde{p}_{\boldsymbol{\alpha}}(\mathbf{z}_0|\mathbf{z}_t,\mathbf{y})$ and the true posterior $q_{star}(\mathbf{z}_0|\mathbf{z}_t,\mathbf{y})$ can be decomposed into two terms:*

$$D_{\mathrm{KL}}\left(\tilde{p}_{\boldsymbol{\alpha}}(\mathbf{z}_0|\mathbf{z}_t,\mathbf{y})\|q(\mathbf{z}_0|\mathbf{z}_t,\mathbf{y})\right)=D_{\mathrm{KL}}\left(\tilde{p}_{\boldsymbol{\alpha}}(\mathbf{z}_0|\mathbf{z}_t,\mathbf{y})\|q_{star}(\mathbf{z}_0|\mathbf{z}_t)\right)-\mathbb{E}_{\mathbf{z}_0\sim\tilde{p}_{\boldsymbol{\alpha}}(\mathbf{z}_0|\mathbf{z}_t,\mathbf{y})}\left[\log q_{star}(\mathbf{y}|\mathbf{z}_0)\right], \tag{8}$$

*Proof.* We begin by considering the KL divergence between the variational distribution $\tilde{p}_{\boldsymbol{\alpha}}(\mathbf{z}_0|\mathbf{z}_t,\mathbf{y})$ and the true posterior $q_{\mathrm{star}}(\mathbf{z}_0|\mathbf{z}_t,\mathbf{y})$. Given that $\mathbf{z}_0$ is a discrete variable, the KL divergence can be expressed as a sum:

$$D_{\mathrm{KL}}\left(\tilde{p}_{\boldsymbol{\alpha}}(\mathbf{z}_0|\mathbf{z}_t,\mathbf{y})\|q_{\mathrm{star}}(\mathbf{z}_0|\mathbf{z}_t,\mathbf{y})\right)=\sum_{\mathbf{z}_0}\tilde{p}_{\boldsymbol{\alpha}}(\mathbf{z}_0|\mathbf{z}_t,\mathbf{y})\log\frac{\tilde{p}_{\boldsymbol{\alpha}}(\mathbf{z}_0|\mathbf{z}_t,\mathbf{y})}{q_{\mathrm{star}}(\mathbf{z}_0|\mathbf{z}_t,\mathbf{y})}. \tag{21}$$

By applying Bayes' theorem to the true posterior $q_{\mathrm{star}}(\mathbf{z}_0|\mathbf{z}_t,\mathbf{y})$, we have:

$$q_{\mathrm{star}}(\mathbf{z}_0|\mathbf{z}_t,\mathbf{y})=\frac{q_{\mathrm{star}}(\mathbf{z}_0|\mathbf{z}_t)q_{\mathrm{star}}(\mathbf{y}|\mathbf{z}_0)}{q_{\mathrm{star}}(\mathbf{y}|\mathbf{z}_t)}. \tag{22}$$

Since $q_{\text{star}}(\mathbf{y}|\mathbf{z}_t)$ does not depend on $\mathbf{z}_0$, it can be treated as a constant and ignored in the KL divergence calculation. Substituting Eq. (22) into Eq. (21), we obtain:

$$D_{\text{KL}}\left(\tilde{p}_{\boldsymbol{\alpha}}(\mathbf{z}_0|\mathbf{z}_t,\mathbf{y})\|q_{\text{star}}(\mathbf{z}_0|\mathbf{z}_t,\mathbf{y})\right) = \sum_{\mathbf{z}_0} \tilde{p}_{\boldsymbol{\alpha}}(\mathbf{z}_0|\mathbf{z}_t,\mathbf{y}) \log \frac{\tilde{p}_{\boldsymbol{\alpha}}(\mathbf{z}_0|\mathbf{z}_t,\mathbf{y})}{q_{\text{star}}(\mathbf{z}_0|\mathbf{z}_t)q_{\text{star}}(\mathbf{y}|\mathbf{z}_0)}. \tag{23}$$

Next, we split the logarithm in the numerator and denominator:

$$D_{\text{KL}}\left(\tilde{p}_{\boldsymbol{\alpha}}(\mathbf{z}_0|\mathbf{z}_t,\mathbf{y})\|q_{\text{star}}(\mathbf{z}_0|\mathbf{z}_t,\mathbf{y})\right) = \sum_{\mathbf{z}_0} \tilde{p}_{\boldsymbol{\alpha}}(\mathbf{z}_0|\mathbf{z}_t,\mathbf{y}) \left[\log \frac{\tilde{p}_{\boldsymbol{\alpha}}(\mathbf{z}_0|\mathbf{z}_t,\mathbf{y})}{q_{\text{star}}(\mathbf{z}_0|\mathbf{z}_t)} - \log q_{\text{star}}(\mathbf{y}|\mathbf{z}_0)\right]. \tag{24}$$

This expression can be decomposed into two terms:

1. The first term represents the KL divergence between the variational distribution $\tilde{p}_{\boldsymbol{\alpha}}(\mathbf{z}_0|\mathbf{z}_t,\mathbf{y})$ and the prior $q_{\text{star}}(\mathbf{z}_0|\mathbf{z}_t)$:

$$D_{\text{KL}}\left(\tilde{p}_{\boldsymbol{\alpha}}(\mathbf{z}_0|\mathbf{z}_t,\mathbf{y})\|q_{\text{star}}(\mathbf{z}_0|\mathbf{z}_t)\right) = \sum_{\mathbf{z}_0} \tilde{p}_{\boldsymbol{\alpha}}(\mathbf{z}_0|\mathbf{z}_t,\mathbf{y}) \log \frac{\tilde{p}_{\boldsymbol{\alpha}}(\mathbf{z}_0|\mathbf{z}_t,\mathbf{y})}{q_{\text{star}}(\mathbf{z}_0|\mathbf{z}_t)}. \tag{25}$$

2. The second term is the negative expected log-likelihood under the variational distribution:

$$\mathbb{E}_{\mathbf{z}_0 \sim \tilde{p}_{\boldsymbol{\alpha}}(\mathbf{z}_0|\mathbf{z}_t,\mathbf{y})}\left[-\log q_{\text{star}}(\mathbf{y}|\mathbf{z}_0)\right] = -\sum_{\mathbf{z}_0} \tilde{p}_{\boldsymbol{\alpha}}(\mathbf{z}_0|\mathbf{z}_t,\mathbf{y}) \log q_{\text{star}}(\mathbf{y}|\mathbf{z}_0). \tag{26}$$

Thus, the KL divergence between $\tilde{p}_{\boldsymbol{\alpha}}(\mathbf{z}_0|\mathbf{z}_t,\mathbf{y})$ and $q_{\text{star}}(\mathbf{z}_0|\mathbf{z}_t,\mathbf{y})$ can be decomposed as follows:

$$D_{\text{KL}}\left(\tilde{p}_{\boldsymbol{\alpha}}(\mathbf{z}_0|\mathbf{z}_t,\mathbf{y})\|q_{\text{star}}(\mathbf{z}_0|\mathbf{z}_t,\mathbf{y})\right) = D_{\text{KL}}\left(\tilde{p}_{\boldsymbol{\alpha}}(\mathbf{z}_0|\mathbf{z}_t,\mathbf{y})\|q_{\text{star}}(\mathbf{z}_0|\mathbf{z}_t)\right) - \mathbb{E}_{\mathbf{z}_0 \sim \tilde{p}_{\boldsymbol{\alpha}}(\mathbf{z}_0|\mathbf{z}_t,\mathbf{y})}\left[\log q_{\text{star}}(\mathbf{y}|\mathbf{z}_0)\right]. \tag{27}$$

This concludes the proof. $\qquad\square$

**Lemma D.1.** *The marginal distribution $q_{star\text{-}decomp}(\mathbf{z}_0|\mathbf{y})$ is identical to the target distribution $q_{star}(\mathbf{z}_0|\mathbf{y})$.*

*Proof.* We aim to show that

$$q_{\text{star-decomp}}(\mathbf{z}_0|\mathbf{y}) = q_{\text{star}}(\mathbf{z}_0|\mathbf{y}). \tag{28}$$

Recall first that $q_{\text{star}}(\mathbf{z}_{0:T}|\mathbf{y})$ is our original "star-shaped" joint posterior, and we define its "star-decomposed" variant by

$$q_{\text{star-decomp}}(\mathbf{z}_{0:T}|\mathbf{y}) = q_{\text{star}}(\mathbf{z}_T|\mathbf{y}) \prod_{t=1}^{T} q_{\text{star}}(\mathbf{z}_{t-1}|\mathbf{z}_t,\mathbf{y}). \tag{29}$$

By the definition of marginalization, to get $q_{\text{star-decomp}}(\mathbf{z}_0|\mathbf{y})$, we integrate over all the latent variables $\mathbf{z}_1,\ldots,\mathbf{z}_T$:

$$q_{\text{star-decomp}}(\mathbf{z}_0|\mathbf{y}) = \sum_{\mathbf{z}_{1:T}} q_{\text{star-decomp}}(\mathbf{z}_{0:T}|\mathbf{y}). \tag{30}$$

Substituting our definition of $q_{\text{star-decomp}}(\mathbf{z}_{0:T}|\mathbf{y})$ into the sum yields

$$q_{\text{star-decomp}}(\mathbf{z}_0|\mathbf{y}) = \sum_{\mathbf{z}_{1:T}} \left[q_{\text{star}}(\mathbf{z}_T|\mathbf{y}) \prod_{t=1}^{T} q_{\text{star}}(\mathbf{z}_{t-1}|\mathbf{z}_t,\mathbf{y})\right]. \tag{31}$$

Here, $q_{\text{star}}(\mathbf{z}_T|\mathbf{y})$ and each factor $q_{\text{star}}(\mathbf{z}_{t-1}|\mathbf{z}_t, \mathbf{y})$ are just conditional distributions of the star-shaped model. Next, we marginalize over $\mathbf{z}_T$, then $\mathbf{z}_{T-1}$, and so on, one index at a time. That is, we view

$$\sum_{\mathbf{z}_T} q_{\text{star}}(\mathbf{z}_T|\mathbf{y})\, q_{\text{star}}(\mathbf{z}_{T-1}|\mathbf{z}_T, \mathbf{y}) = q_{\text{star}}(\mathbf{z}_{T-1}|\mathbf{y}),$$

because summation over $\mathbf{z}_T$ of $q_{\text{star}}(\mathbf{z}_T|\mathbf{y})\, q_{\text{star}}(\mathbf{z}_{T-1}|\mathbf{z}_T, \mathbf{y})$ collapses exactly to the marginal $q_{\text{star}}(\mathbf{z}_{T-1}|\mathbf{y})$. This step follows directly from the chain rule of probability (or the law of total probability).

We then apply the same idea repeatedly: first summing out $\mathbf{z}_T$, which gives us a factor depending only on $\mathbf{z}_{T-1}$; then summing out $\mathbf{z}_{T-1}$ to obtain a factor depending on $\mathbf{z}_{T-2}$; and so on, down through $\mathbf{z}_1$. Performing these consecutive sums from $t = T$ down to $t = 1$ eventually leaves only $\mathbf{z}_0$ in the expression. Hence, we get

$$q_{\text{star-decomp}}(\mathbf{z}_0|\mathbf{y}) =\ q_{\text{star}}(\mathbf{z}_0|\mathbf{y}), \tag{32}$$

which shows that both star-decomposed and original star-shaped posteriors yield the same marginal distribution over $\mathbf{z}_0$. □

# E    Details on Experiments

## E.1    Image inverse problems

**Implementation of Forward Operators and Dataset Selection**    In our image inverse problem experiments, the definition and implementation of the forward operator are based on the DPS implementation[2]. To ensure a diverse representation of ImageNet classes without genre bias, we select 1000 images covering all classes $0, 1, 2, \ldots, 999$ using the `imagenet_val_1k.txt` file provided by Pan et al. (2021)[3]. For our experiments with the FFHQ dataset, we use the 1000 images (indexed $0, 1, \ldots, 999$) from the validation set.

## E.2    Implementation Details of G2D2 in Inverse Problem Settings

The implementation of G2D2 is based on the VQ-Diffusion model from the `diffusers` library [4]. For the prior model, we use the pre-trained model available at `https://huggingface.co/microsoft/vq-diffusion-ithq`. In our experiments, the number of time steps $T$ for sampling is set to 100.

**Parameterization of Star-Shaped Noise Process**    In G2D2, the star-shaped noise process follows the same cumulative transition probability $q(\mathbf{z}_t|\mathbf{z}_0)$ as the original Markov noise process. For the Markov noise forward process where $q(\mathbf{z}_t|\mathbf{z}_{t-1})$ is defined using $Q_t$ as in Equation 2, the cumulative transition probability is computed as $q(z_{t,i}|\mathbf{z}_0) = \boldsymbol{v}^{\mathsf{T}}(z_{t,i})\overline{Q}_t\boldsymbol{v}(z_{0,i})$, where $\overline{Q}_t = Q_t \cdots Q_1$. Here, $\overline{Q}_t$ can be computed in closed form as:

$$\overline{Q}_t\boldsymbol{v}(z_{0,i}) = \overline{\alpha}_t\boldsymbol{v}(z_{0,i}) + (\overline{\gamma}_t - \overline{\beta}_t)\boldsymbol{v}(K+1) + \overline{\beta}_t, \tag{33}$$

where $\overline{\alpha}_t = \prod_{i=1}^{t-1}\alpha_i$, $\overline{\gamma}_t = 1 - \prod_{i=1}^{t-1}(1 - \gamma_i)$, and $\overline{\beta}_t = (1 - \overline{\alpha}_t - \overline{\gamma}_t)/(K+1)$. These parameters can be calculated and stored in advance. The parameter settings follow those used during the training of the prior model. Specifically, $\overline{\alpha}_1$ is set to 0.99999, $\overline{\alpha}_T$ to 0.000009, $\overline{\gamma}_1$ to 0.000009, and $\overline{\gamma}_T$ to 0.99999. For both $\overline{\alpha}_t$ and $\overline{\gamma}_t$, values are linearly interpolated between steps 1 and $T$. This scheduling results in a linear increase in the number of [MASK] states as $t$ increases, ultimately leading to all variables transitioning to the [MASK] state. Additionally, the transition probability $\beta_t$ between unmasked tokens is set to be negligibly small, as $\overline{\alpha}_t$ and $\overline{\gamma}_t$ sum to nearly 1.

---

[2] `https://github.com/DPS2022/diffusion-posterior-sampling`
[3] `https://github.com/XingangPan/deep-generative-prior/`
[4] `https://huggingface.co/docs/diffusers/main/en/api/pipelines/vq_diffusion`

**Optimization in the Algorithm and Instantiation of the Objective Function**   In the continuous optimization phase, we optimize the parameters $\boldsymbol{\alpha}$ of the categorical distribution using the RAdam optimizer (Liu et al., 2020). The optimization objective is a weighted sum of the KL divergence term and the likelihood term, defined as:

$$\boldsymbol{\alpha}_t = \arg\min_{\boldsymbol{\alpha}_t} \left\{ \eta_{\mathrm{KL}} D_{\mathrm{KL}} \left( \tilde{p}_{\boldsymbol{\alpha}}(\mathbf{z}_0|\mathbf{z}_t, \mathbf{y}) \| \tilde{p}_\theta(\mathbf{z}_0|\mathbf{z}_t) \right) + \| \mathbf{y} - \mathbf{A}\mathbf{x}_0(\boldsymbol{\alpha}_t) \|_2 \right\}, \tag{34}$$

where $\eta_{\mathrm{KL}}$ controls the trade-off between the KL term and the likelihood term.

**Marginalization over $\mathbf{z}_0$ in Algorithm 1**   The marginalization over $\mathbf{z}_0$ in line 10 of Algorithm 1, specifically the term $\sum_{\mathbf{z}_0} q_{\mathrm{star}}(\mathbf{z}_{t-1}|\mathbf{z}_0) \tilde{p}_{\boldsymbol{\alpha}}(\mathbf{z}_0|\mathbf{z}_t, \mathbf{y})$, can be computed in closed form. This computation is feasible because both distributions involved in the marginalization are dimensionally independent categorical distributions, as discussed by Austin et al. (2021) and Gu et al. (2022).

**Dynamic Learning Rate and KL Coefficient Scheduling**   Some parameters are dynamically adjusted during inference. Both the learning rate for RAdam ($l_{\mathrm{RAdam}}$) and the KL divergence coefficient ($\eta_{\mathrm{KL}}$) are scheduled using weight vectors that decay logarithmically over the inference steps. These weights are computed based on initial scaling factors.

The learning rate weight vector $w_{\mathrm{lr}}$ and the KL coefficient weight vector $w_{\mathrm{KL}}$ are defined as follows:

$$w_{\mathrm{lr}}(t) = 10^{\left( \frac{\lambda_{\mathrm{lr,\ schedule}}}{2} \cdot \left( \frac{2t}{T} - 1 \right) \right)},$$

$$w_{\mathrm{KL}}(t) = 10^{\left( \frac{\lambda_{\mathrm{KL,\ schedule}}}{2} \cdot \left( \frac{2t}{T} - 1 \right) \right)}.$$

Here, $\lambda_{\mathrm{lr,\ schedule}}$ and $\lambda_{\mathrm{KL,\ schedule}}$ represent the initial scaling factors for the learning rate and KL coefficient, respectively, and $T$ is the total number of inference steps. When $\lambda_{\mathrm{lr,\ schedule}} > 0$, the learning rate weight vector $w_{\mathrm{lr}}(t)$ starts with relatively large values when $t$ is large and decays exponentially as $t$ decreases. Specifically, $w_{\mathrm{lr}}(t)$ reaches its minimum near $t = 1$ and its maximum near $t = T$. This scheduling enables stronger optimization during the initial inference steps, with the learning rate gradually decreasing in the later steps.

At each step $t$, the parameters are set as follows:

$$l_{\mathrm{RAdam}}(t) = l_{\mathrm{RAdam,\ base}} \cdot w_{\mathrm{lr}}(t), \quad \eta_{\mathrm{KL}}(t) = \eta_{\mathrm{KL,\ base}} \cdot w_{\mathrm{KL}}(t).$$

**Task-Specific and Common Hyperparameters**   The hyperparameters for Gaussian deblurring and super-resolution tasks used in the experiments are shown in Table 4.

The following hyperparameters are shared across all experiments: The number of iterations for the optimization is set to 30, the temperature for Gumbel-Softmax relaxation is 1.0, and the forget coefficient is 0.3. For the classifier-free guidance scale, we use 5.0 in ImageNet experiments and 3.0 in FFHQ experiments.

### E.3   G2D2 with Markov Noise Process

As discussed in Section 4.3, a variant of G2D2 can be derived by introducing the original Markov noise process in the graphical model. In that case, the algorithm is shown in Algorithm 2. The key point here is that the $q_{\mathrm{Markov}}(\mathbf{z}_{t-1}|\mathbf{z}_0, \mathbf{z}_t)$ part is identical to that of the original Markov noise process, which is expressed as

$$q_{\mathrm{Markov}}(z_{t-1,i}|\mathbf{z}_0, \mathbf{z}_t) = \frac{(\boldsymbol{v}^{\mathsf{T}}(z_{t,i}) Q_t \boldsymbol{v}(z_{t-1,i}))(\boldsymbol{v}^{\mathsf{T}}(z_{t-1,i}) \overline{Q}_{t-1} \boldsymbol{v}(z_{0,i}))}{\boldsymbol{v}^{\mathsf{T}}(z_{t,i}) \overline{Q}_t \boldsymbol{v}(z_{0,i})}. \tag{35}$$

In the mask-absorbing type of Markov noise process, this posterior distribution does not revert tokens that have once become unmasked states back to masked tokens. As a result, it becomes difficult to correct errors that occur in the early stages of sampling in subsequent steps.

| Dataset | Task | Hyperparameter | Value |
|---|---|---|---|
| ImageNet | Gaussian Deblurring | $\eta_{\text{KL, base}}$ | 0.0003 |
| | | $\lambda_{\text{KL, schedule}}$ | 2.0 |
| | | $l_{\text{RAdam, base}}$ | 15.0 |
| | | $\lambda_{\text{lr, schedule}}$ | 1.0 |
| ImageNet | Super-resolution | $\eta_{\text{KL, base}}$ | 0.0003 |
| | | $\lambda_{\text{KL, schedule}}$ | 2.0 |
| | | $l_{\text{RAdam, base}}$ | 10.0 |
| | | $\lambda_{\text{lr, schedule}}$ | 1.0 |
| FFHQ | Gaussian Deblurring | $\eta_{\text{KL, base}}$ | 0.0003 |
| | | $\lambda_{\text{KL, schedule}}$ | 2.0 |
| | | $l_{\text{RAdam, base}}$ | 15.0 |
| | | $\lambda_{\text{lr, schedule}}$ | 1.0 |
| FFHQ | Super-resolution | $\eta_{\text{KL, base}}$ | 0.0003 |
| | | $\lambda_{\text{KL, schedule}}$ | 2.0 |
| | | $l_{\text{RAdam, base}}$ | 10.0 |
| | | $\lambda_{\text{lr, schedule}}$ | 2.0 |

Table 4: Hyperparameters for Gaussian Deblurring and Super-resolution tasks on ImageNet and FFHQ datasets.

We compare the computational complexity of G2D2 with the star-shaped noise process (Algorithm 1) and its hypothetical Markovian counterpart (Algorithm 2). The main algorithmic difference lies in line 10 of both algorithms: the sampling of $\mathbf{z}_{t-1}$ given the optimized $\tilde{p}_{\boldsymbol{\alpha}}(\mathbf{z}_0|\mathbf{z}_t, \mathbf{y})$.

The key differences between both algorithms are as follows. In G2D2 (Algorithm 1), we have $p_{\alpha}(\mathbf{z}_{t-1}|\mathbf{z}_t, \mathbf{y}) = \sum_{\mathbf{z}_0} q_{\text{star}}(\mathbf{z}_{t-1}|\mathbf{z}_0)\tilde{p}_{\boldsymbol{\alpha}}(\mathbf{z}_0|\mathbf{z}_t, \mathbf{y})$, where $q_{\text{star}}(\mathbf{z}_{t-1}|\mathbf{z}_0)$ represents the direct transition from $\mathbf{z}_0$ to $\mathbf{z}_{t-1}$ based on the cumulative transition matrix $\bar{Q}_{t-1}$ (derived from $Q_1, \ldots, Q_{t-1}$ as described in Appendix E.2). In the Markovian Version (Algorithm 2), we have $p_{\alpha}(\mathbf{z}_{t-1}|\mathbf{z}_t, \mathbf{y}) = \sum_{\mathbf{z}_0} q_{\text{Markov}}(\mathbf{z}_{t-1}|\mathbf{z}_0, \mathbf{z}_t)\tilde{p}_{\boldsymbol{\alpha}}(\mathbf{z}_0|\mathbf{z}_t, \mathbf{y})$, where $q_{\text{Markov}}(\mathbf{z}_{t-1}|\mathbf{z}_0, \mathbf{z}_t)$ is the posterior from the original Markov process in (35).

Crucially, the computational cost of sampling $\mathbf{z}_{t-1}$ in both cases is dominated by the marginalization over $\mathbf{z}_0$ after $\tilde{p}_{\boldsymbol{\alpha}}(\mathbf{z}_0|\mathbf{z}_t, \mathbf{y})$ has been optimized. The calculation of $q_{\text{star}}(\mathbf{z}_{t-1}|\mathbf{z}_0)$ and $q_{\text{Markov}}(\mathbf{z}_{t-1}|\mathbf{z}_0, \mathbf{z}_t)$ involves operations on categorical distributions and transition matrices, which are generally efficient and do not introduce significant computational differences between the two approaches.

---

**Algorithm 2 G2D2 with Markov Noise Process**

---

**Require:** Input condition $\mathbf{y}$, pre-trained discrete diffusion model $p_\theta$, forget coefficient $\gamma$
1: $\mathbf{z}_T \sim q(\mathbf{z}_T)$
2: **for** $t = T, \ldots, 1$ **do**
3:     **if** $t = T$ **then**
4:         Initialize: $\boldsymbol{\alpha}_t \propto \log \tilde{p}_\theta(\mathbf{z}_0|\mathbf{z}_t)$
5:     **else**
6:         Initialize: $\boldsymbol{\alpha}_t \propto \exp(\gamma \log \boldsymbol{\alpha}_{t+1} + (1 - \gamma) \log \tilde{p}_\theta(\mathbf{z}_0|\mathbf{z}_t))$
7:     **end if**
8:     `// continuous optimization`
9:     $\boldsymbol{\alpha}_t = \arg\min_{\boldsymbol{\alpha}_t} D_{\text{KL}}\left(\tilde{p}_{\boldsymbol{\alpha}}(\mathbf{z}_0|\mathbf{z}_t, \mathbf{y}) \| \tilde{p}_\theta(\mathbf{z}_0|\mathbf{z}_t)\right) - \mathbb{E}_{\mathbf{z}_0 \sim \tilde{p}_{\boldsymbol{\alpha}}(\mathbf{z}_0|\mathbf{z}_t, \mathbf{y})}\left[\log q(\mathbf{y}|\mathbf{z}_0)\right]$
10:     Sample $\mathbf{z}_{t-1} \sim p_{\alpha}(\mathbf{z}_{t-1}|\mathbf{z}_t, \mathbf{y}) = \sum_{\mathbf{z}_0} q_{\text{Markov}}(\mathbf{z}_{t-1}|\mathbf{z}_0, \mathbf{z}_t)\tilde{p}_{\boldsymbol{\alpha}}(\mathbf{z}_0|\mathbf{z}_t, \mathbf{y})$
11:     `// Note: The term` $q_{\text{Markov}}(\mathbf{z}_{t-1}|\mathbf{z}_0, \mathbf{z}_t)$ `uses the posterior distribution of the original Markov noise process.`
12: **end for**
13: **return** $\mathbf{x}_0$ by decoding $\mathbf{z}_0$

---

### E.4 Settings for comparison methods

In this subsection, we detail the experimental settings for the comparison methods.

**DPS** (Chung et al., 2023b) We use the same parameter settings as described in the original paper. The guidance scale is set to 1.0 for FFHQ & super-resolution, 1.0 for FFHQ & Gaussian deblurring, 1.0 for ImageNet & super-resolution, and 0.4 for ImageNet & Gaussian deblurring. The number of time steps is set to 1000. For pre-trained models, we use the unconditional model provided by Dhariwal & Nichol (2021) [5] for ImageNet. For FFHQ, we use the model provided by Choi et al. (2021) [6].

**DDRM** (Kawar et al., 2022) We use the official implementation [7]. The time steps are set to $T = 20$, with $\eta = 0.85$ and $\eta_b = 1.0$ as the hyperparameters. For ImageNet, we use the same pre-trained model as DPS. Although there is no official implementation using a pre-trained model trained on FFHQ, both DDRM and Choi et al. (2021) are based on the implementation of Dhariwal & Nichol (2021). Therefore, in our experiments, DDRM uses the same pre-trained model as DPS.

**PSLD** (Rout et al., 2023) We use the official implementation [8]. For the pre-trained model, we employ stable-diffusion v-1.5 (Rombach et al., 2022) [9]. As this model handles 512×512 pixel images, we first upscale the ground truth image to 512×512. We then apply the forward operator to the upscaled image and use the result as observed data for our method. Finally, we downsample the output to 256×256. For hyperparameters, we use $\eta = 1.0$ and $\gamma = 0.1$.

**ReSample** (Song et al., 2024) We use the official implementation [10]. For pre-trained models, we employ two models from the latent diffusion models repository [11]: LDM-VQ-4 trained on FFHQ, and LDM-VQ-8 trained on ImageNet with class conditioning. We use $T = 500$ DDIM steps with $\tau$ set to $10^{-4}$. The maximum number of optimization steps is set to 500. The variance hyperparameter $\gamma$ is set to 40. For the ImageNet experiments, we input the class labels of the ground truth data to the model.

### E.5 GPU memory usage and computational speed

We analyze the GPU memory consumption and computational speed of our proposed method, G2D2, in comparison with other methods. Table 5 presents an overview of these metrics for various methods. The measurements are conducted using a single NVIDIA A6000 GPU for the Gaussian deblurring task on ImageNet. G2D2 has the lowest memory usage among all methods and the fastest computational speed among gradient-based methods.

Table 5: Comparison of GPU Memory Usage and Computational Speed

| Method | GPU Memory Usage (GiB) | Wall-Clock time (s) |
|---|---|---|
| G2D2 (Proposed) | 4.7 | 194 |
| DPS | 10.7 | 277 |
| DDRM | 5.8 | 4 |
| PSLD | 20.9 | 738 |
| ReSample | 7.1 | 555 |

---

[5]https://github.com/openai/guided-diffusion

[6]https://github.com/jychoi118/ilvr_adm

[7]https://github.com/bahjat-kawar/ddrm

[8]https://github.com/LituRout/PSLD

[9]https://github.com/CompVis/stable-diffusion

[10]https://github.com/soominkwon/resample

[11]https://github.com/CompVis/latent-diffusion

### E.6   Impact of the Forget Coefficient

Figure 6 shows the reduction in the loss function and the final results for the Gaussian deblurring task on ImageNet when the forget coefficient is set to 0.3 and 1.0. The case with a forget coefficient of 1.0 corresponds to not using the optimization results from the previous step at all. Introducing the forget coefficient allows for a faster reduction in the loss function and achieves higher performance with the same computational resources.

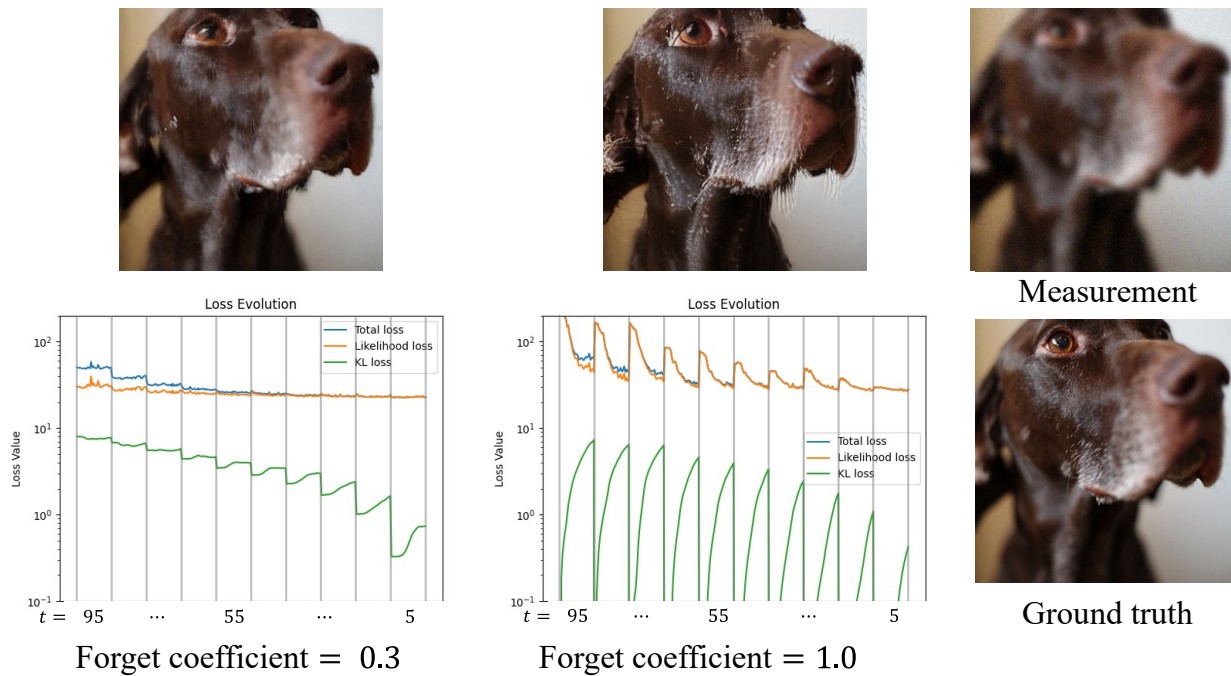

Figure 6: Reduction in the loss function and final results for the Gaussian deblurring task on ImageNet with forget coefficients of 0.3 and 1.0. The forget coefficient of 1.0 corresponds to not using the optimization results from the previous step.

### E.7   Impact of Text Conditioning on the Prior Model

To examine the necessity of text conditioning, we investigate the effect of the presence or absence of prompts given to VQ-Diffusion on performance. Table 6 shows the performance for each setting. "Not Used" for text conditioning indicates that classifier-free guidance in the prior model is set to 1.0 (equivalent to unconditional sampling). The prompts we provide to VQ-Diffusion in our method are "a photo of [Class Name]" for ImageNet experiments and "a high-quality headshot of a person" for FFHQ experiments. It should be noted that these prompts are extremely general and do not describe specific details of the images.

From these results, we can confirm that prompt conditioning contributes to a certain level of performance improvement on ImageNet, while it does not significantly affect performance on FFHQ. This suggests that the pre-trained VQ-Diffusion model may not have been extensively trained on human face images, or that our chosen prompts for FFHQ may not be optimal.

Additionally, Figure 7 shows a qualitative comparison for the Gaussian deblurring task on the ImageNet dataset. When prompts are not used, smoother results are obtained throughout the intermediate steps when compared, demonstrating that the presence of text conditioning leads to final results that capture more fine-grained details.

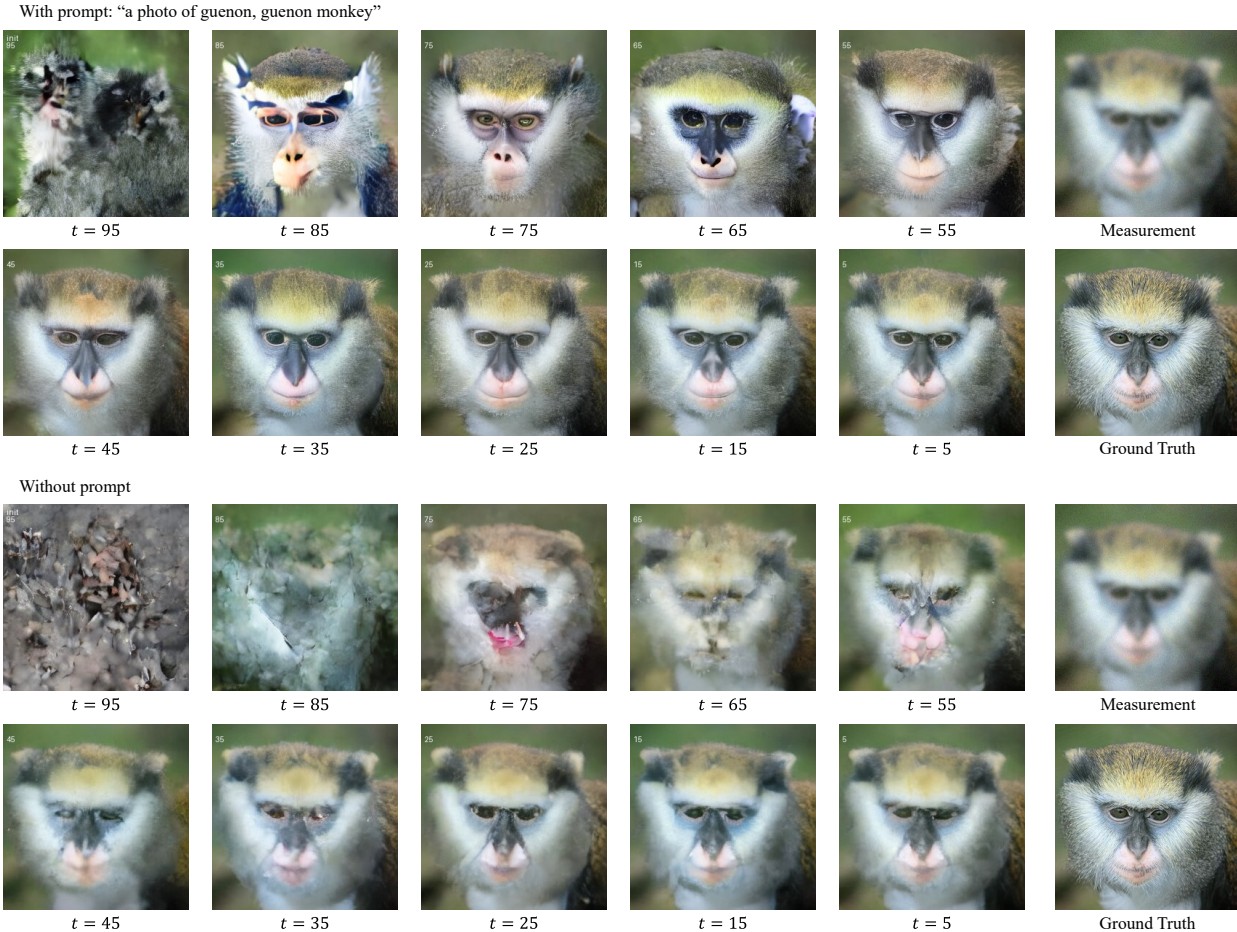

Figure 7: Images sampled from the prior model $\tilde{p}_\theta(\mathbf{z}_0|\mathbf{z}_t)$ using intermediate $\mathbf{z}_t$ during the process of G2D2 in image inverse problem solving. The top two rows show intermediate samples using the prompt "a photo of guenon, guenon monkey," while the bottom two rows show results without using any prompt. Each column represents a different time step in the G2D2 algorithm, ranging from $t = 95$ (earliest stage) to $t = 5$ (final stage). The rightmost column displays the measurement (degraded input image) and ground truth (original image).

Table 6: Performance comparison with and without text conditioning

| Dataset | Text conditioning | SR (×4) | | Gaussian Deblurring | |
|---|---|---|---|---|---|
| | | LPIPS↓ | PSNR↑ | LPIPS↓ | PSNR↑ |
| ImageNet | Not Used | 0.357 | **23.55** | 0.385 | **23.24** |
| | Used | **0.351** | 23.37 | **0.370** | 23.14 |
| FFHQ | Not Used | **0.258** | **27.48** | 0.274 | 26.99 |
| | Used | 0.259 | 27.44 | **0.273** | **27.00** |

## E.8  Impact of prior selection for G2D2

G2D2 can also use other prior models when solving inverse problems. In this paper, we primarily use VQ-Diffusion trained on the ITHQ dataset (hereafter **VQDiff-ITHQ**) as a prior model, but to investigate how different prior models affect inverse problem solving performance, we conduct experiments using

another model, CLIP-VQDiffusion (Han & Kim, 2024) [12], trained on the FFHQ dataset (hereafter **CVQDiff-FFHQ**). As in the main text, we evaluated on Gaussian deblurring tasks and super-resolution tasks (with measurement noise $\sigma = 0.05$). For evaluation, we used 100 images from the FFHQ validation subset and measured PSNR, LPIPS.

Table 7: Performance comparison between different prior models on FFHQ dataset

| Dataset | Prior Model | SR (×4) | | Gaussian Deblurring | |
|---------|-------------|---------|---------|---------|---------|
| | | PSNR↑ | LPIPS↓ | PSNR↑ | LPIPS↓ |
| FFHQ | **VQDiff-ITHQ** (Tang et al., 2022) | **27.44** | **0.259** | **27.00** | **0.273** |
| | **CVQDiff-FFHQ** (Han & Kim, 2024) | 25.32 | 0.283 | 24.41 | 0.289 |

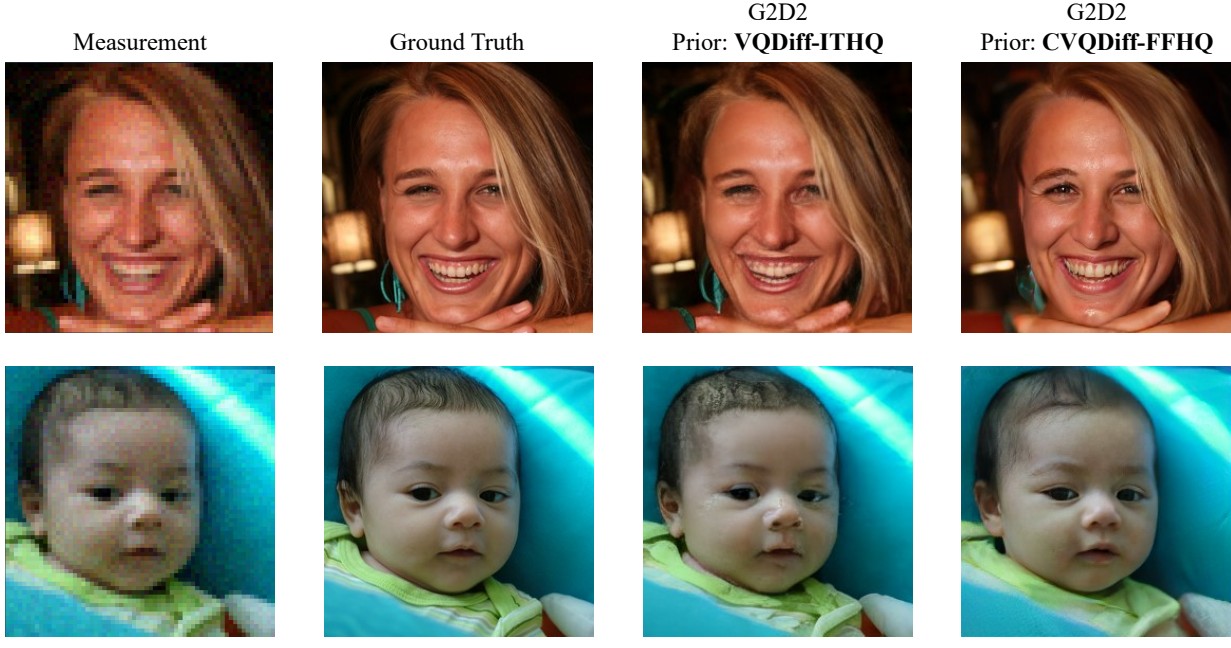

Super resolution, (x4, $\sigma = 0.05$)

Figure 8: Qualitative comparison of super-resolution results (×4, $\sigma = 0.05$) on FFHQ test images. From left to right: low-resolution measurement, ground truth, G2D2 with **VQDiff-ITHQ** prior, and G2D2 with **CVQDiff-FFHQ** prior. While both prior models produce visually pleasing reconstructions, **VQDiff-ITHQ** prior tends to generate results closer to the ground truth, whereas **CVQDiff-FFHQ** prior produces fewer artifacts but with slightly less fidelity to the ground truth images.

Table 7 shows the quantitative results, and Figure 8 presents examples of reconstructed samples. Interestingly, although **CVQDiff-FFHQ** prior is trained on the FFHQ dataset, G2D2 with **VQDiff-ITHQ** prior performs slightly better. When observing qualitative results, G2D2 with **CVQDiff-FFHQ** prior produces fewer artifacts but seems to deviate slightly from the ground truth images. Since **CVQDiff-FFHQ** prior is trained on FFHQ, it likely has an advantage in prior modeling (corresponding to the first term of the optimization target in Algorithm 1, L10). However, the optimization of the likelihood in the measurement equation (the second term) involves decoder calculations, and since both models use different decoders, this difference may be influencing the results.

---

[12]https://github.com/INFINIQ-AI1/CLIPVQDiffusion

### E.9 Failure modes of G2D2

We conduct an analysis of failure modes. Figure 9 shows the results of G2D2 and the images during inference for the Gaussian deblurring task on FFHQ. When the ground truth image is a relatively young (child's) face, the generated face images appear to be drawn towards a distribution of more adult faces. This is likely due to the use of the prompt "a high-quality headshot of a person". As a result, there is a consistent bias towards adult face images throughout the generation process, leading to artifacts in the final image. In the absence of a prompt, the intermediate generated images are not influenced by any specific textual guidance. As a result, the final image tends to have fewer artifacts.

While the star-shaped noise process can correct early errors, if errors persist until the later stages, it becomes more difficult to correct them from that point onwards. In other words, when there is a mismatch between the distribution conditioned by the prompt and the target image, it becomes challenging for G2D2 to handle it effectively.

To improve these issues, techniques such as simultaneous optimization of prompts may be necessary. Prompt-tuning techniques, as proposed in reference (Chung et al., 2024), could be effective in addressing these challenges.

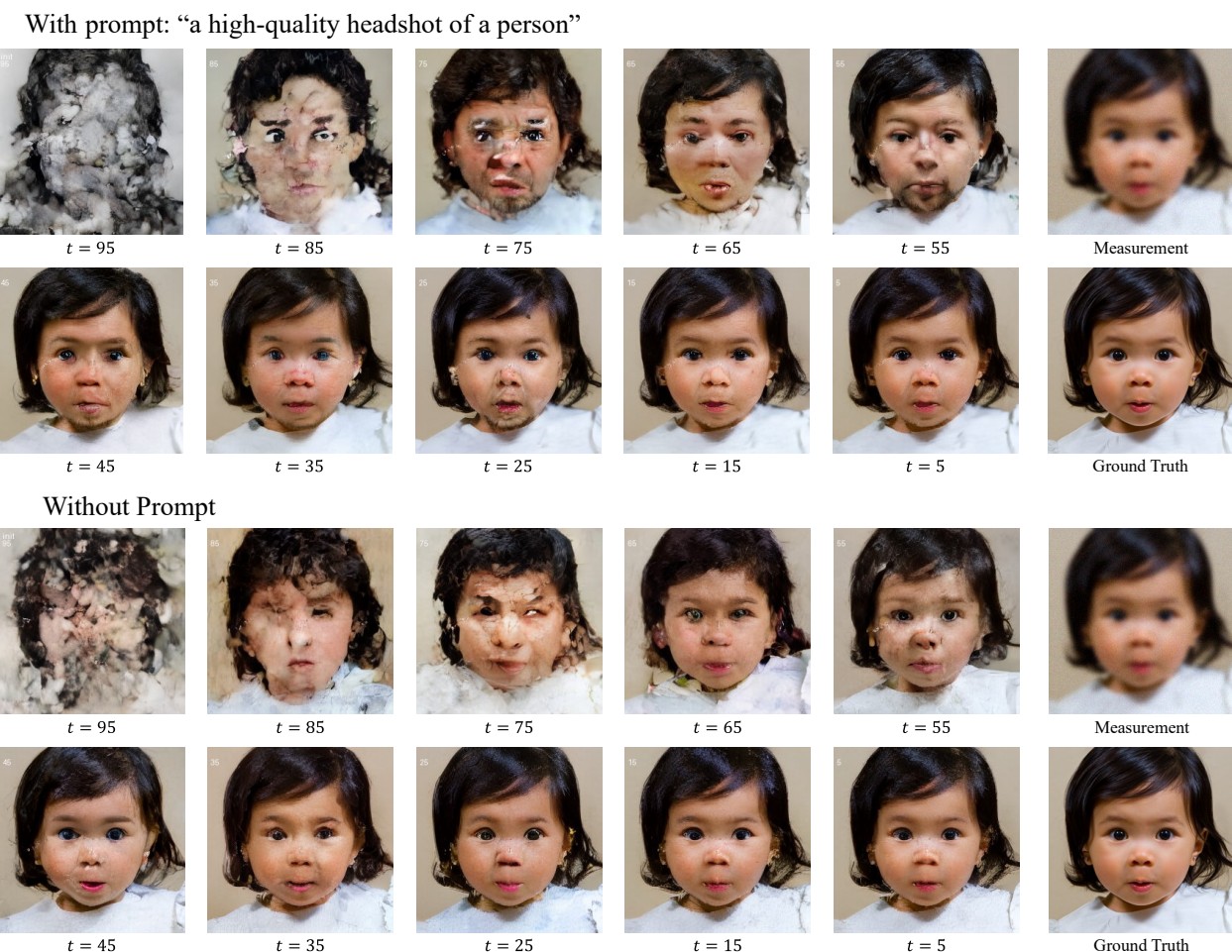

Figure 9: Failure modes of G2D2: Gaussian Deblurring results on the FFHQ dataset. Due to the mismatch between the prompt and the target image, errors remain uncorrected throughout the process, resulting in artifacts in the estimated image.

### E.10 Additional quantitative results of G2D2 and comparison methods.

Tables 8-11 shows the evaluation results with SSIM, FID, and KID added as evaluation metrics.

Tables 8–11 show the evaluation results with Structural Similarity Index Measure (SSIM) (Wang et al., 2004), Fréchet Inception Distance (FID) (Heusel et al., 2017), and Kernel Inception Distance (KID) (Bińkowski et al., 2018) added as evaluation metrics. The evaluation of FID and KID in this study is conducted under the following conditions: We use the `clean-fid`[13] library for our evaluation. Following the default settings of the `clean-fid` library, we use 2048-dimensional features extracted from the `pool_3` layer of the Inception V3 model. We follow the internal processing of the `clean-fid` library for image preprocessing, which typically includes resizing input images to 299×299 pixels and appropriate normalization. For reference statistics, we use pre-computed statistics of the `trainval` split of the FFHQ dataset (resolution 256×256, based on 50,000 images) provided by the `clean-fid` library (specified as `dataset_name="ffhq", dataset_res=256, dataset_split="trainval"`). For the ImageNet dataset (256×256 resolution), it should be noted that the reference set for FID calculation on ImageNet consists of 1,000 images, which is smaller than the recommended sample size for typical FID evaluation (e.g., 10,000-50,000 images). Therefore, FID scores under these conditions should be interpreted with consideration of the limitations regarding the reference set size.

Table 8: Quantitative evaluation on ImageNet 256×256. Performance comparison of different methods on Super-Resolution (×4) task. Values show mean over 1000 images.

| Prior Type | Method | PSNR($\uparrow$) | LPIPS($\downarrow$) | SSIM($\uparrow$) | FID($\downarrow$) | KID($\downarrow$)($\times 10^{-2}$) |
|---|---|---|---|---|---|---|
| Pixel-domain | DPS | 22.67 | 0.362 | 0.618 | 45.97 | 0.41 |
| | DDRM | 24.27 | 0.351 | 0.669 | 52.26 | 1.16 |
| LDM | PSLD | 24.02 | 0.331 | 0.675 | 47.88 | 0.83 |
| | ReSample | 23.31 | 0.373 | 0.616 | 68.77 | 1.96 |
| Discrete | G2D2 (proposed) | 23.82 | 0.340 | 0.638 | 49.97 | 0.86 |
| | G2D2 w/ Markov noise process | 22.01 | 0.442 | 0.557 | 78.43 | 2.40 |

Table 9: Quantitative evaluation on ImageNet 256×256. Performance comparison of different methods on Gaussian Deblurring task. Values show mean over 1000 images.

| Prior Type | Method | PSNR($\uparrow$) | LPIPS($\downarrow$) | SSIM($\uparrow$) | FID($\downarrow$) | KID($\downarrow$)($\times 10^{-2}$) |
|---|---|---|---|---|---|---|
| Pixel-domain | DPS | 19.49 | 0.432 | 0.473 | 62.04 | 0.86 |
| | DDRM | 27.61 | 0.250 | 0.786 | 34.05 | 0.40 |
| LDM | PSLD | 23.95 | 0.359 | 0.659 | 50.96 | 0.96 |
| | ReSample | 23.07 | 0.425 | 0.586 | 82.57 | 2.44 |
| Discrete | G2D2 (proposed) | 23.37 | 0.367 | 0.604 | 61.60 | 1.39 |
| | G2D2 w/ Markov noise process | 22.36 | 0.424 | 0.551 | 79.29 | 2.43 |

Table 10: Quantitative evaluation on FFHQ 256×256. Performance comparison of different methods on Super-Resolution (×4) task. Values show the mean over 1000 images.

| Prior Type | Method | PSNR($\uparrow$) | LPIPS($\downarrow$) | SSIM($\uparrow$) | FID($\downarrow$) | KID($\downarrow$)($\times 10^{-2}$) |
|---|---|---|---|---|---|---|
| Pixel-domain | DPS | 26.07 | 0.238 | 0.756 | 26.85 | 0.67 |
| | DDRM | 28.09 | 0.252 | 0.804 | 46.19 | 3.00 |
| LDM | PSLD | 27.12 | 0.282 | 0.757 | 37.37 | 2.07 |
| | ReSample | 23.07 | 0.508 | 0.445 | 85.86 | 7.05 |
| Discrete | G2D2 (proposed) | 27.29 | 0.265 | 0.763 | 45.21 | 3.08 |
| | G2D2 w/ Markov noise process | 25.15 | 0.369 | 0.699 | 62.96 | 4.47 |

---

[13] https://github.com/GaParmar/clean-fid

Table 11: Quantitative evaluation on FFHQ 256×256. Performance comparison of different methods on Gaussian Deblurring task. Values show the mean over 1000 images.

| Prior Type | Method | PSNR(↑) | LPIPS(↓) | SSIM(↑) | FID(↓) | KID(↓)(×10⁻²) |
|---|---|---|---|---|---|---|
| Pixel-domain | DPS | 25.47 | 0.234 | 0.731 | 24.10 | 0.40 |
| | DDRM | 30.89 | 0.209 | 0.863 | 42.51 | 2.51 |
| LDM | PSLD | 26.96 | 0.307 | 0.750 | 38.84 | 1.99 |
| | ReSample | 25.91 | 0.336 | 0.642 | 45.26 | 2.85 |
| Discrete | G2D2 (proposed) | 26.91 | 0.280 | 0.743 | 47.70 | 3.46 |
| | G2D2 w/ Markov noise process | 25.96 | 0.340 | 0.708 | 57.73 | 4.29 |

### E.11 Additional qualitative results of G2D2 and comparison methods.

We present additional qualitative results of G2D2 and comparison methods. Figures 10 through 13 showcase the results for super-resolution and Gaussian deblurring tasks on ImageNet and FFHQ datasets.

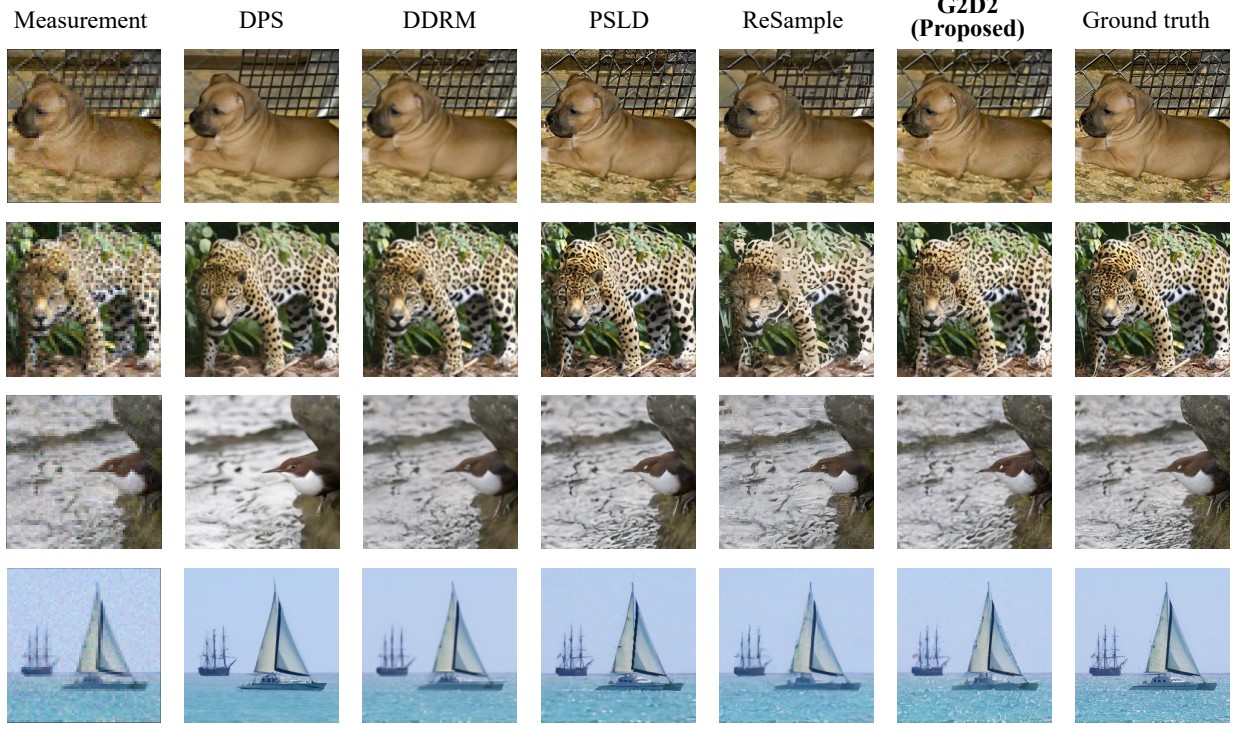

Super resolution (x4, $\sigma = 0.05$)

Figure 10: Qualitative results of G2D2 and comparison methods.

### E.12 Additional qualitative results of G2D2 with Markov noise process

To compare G2D2 and G2D2 with Markov noise process, we present their respective qualitative results in Figures 14 and 15. The latter approach does not include re-masking operations in its sampling process, which means that once a token becomes unmasked, it cannot be modified in subsequent iterations. The unnatural artifacts observed in the resulting images are likely attributable to this limitation. This observation underscores the validity of adopting the star-shaped noise process in our proposed method.

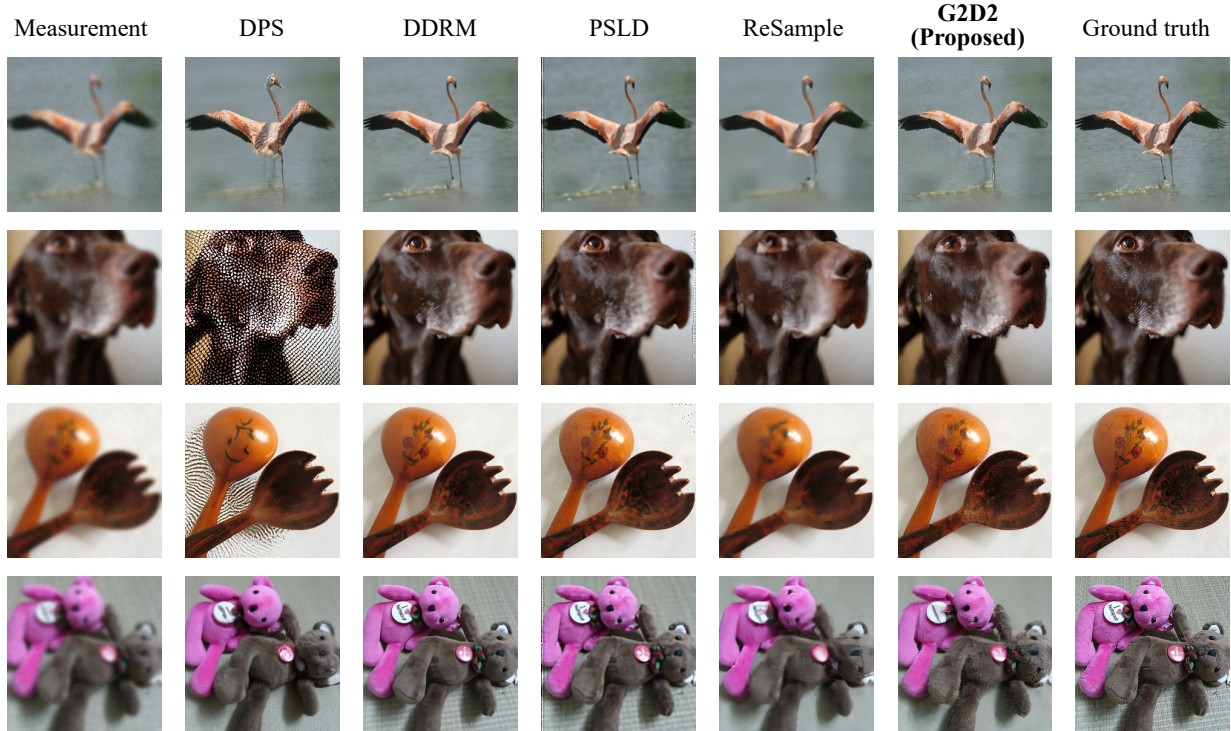

Gaussian Deblur ($\sigma = 0.05$)

Figure 11: Qualitative results of G2D2 and comparison methods.

### E.13 Inverse problems on motion data

We develop G2D2 based on the official implementation of MMM (Pinyoanuntapong et al., 2024) [14]. This method learns a masked generative model on the discrete latent space obtained by a motion tokenizer trained on the VQ-VAE framework (van den Oord et al., 2017). G2D2 uses the provided pre-trained model as a prior distribution.

We conduct experiments on the path following task (Song et al., 2023b; Uchida et al., 2024). The objective is to generate motion data $\mathbf{m}_0 \in \mathbb{R}^{d_{\mathbf{m}} \times L}$ that follows a given path $\mathbf{y}_{\text{path}} \in \mathbb{R}^{3 \times L}$. Here, $\mathbf{y}_{\text{path}}$ represents the coordinates of the hip joint at each time frame, $L$ denotes the number of frames in the motion data, and $d_{\mathbf{m}}$ is the dimensionality of each motion data point.

The likelihood loss used in the optimization process of G2D2 measures how closely the generated motion follows the target path. It is defined as

$$\log q(\mathbf{y}_{\text{path}}|\mathbf{m}_0) = \sum_{l=1}^{L} \|\mathbf{y}_{\text{path},l} - \mathbf{A}_{\text{path}}\mathbf{m}_{0,l}\|_2, \tag{36}$$

where $\mathbf{A}_{\text{path}}$ is a linear operator that extracts the path across the frames.

We conduct experiments with a total of $T = 25$ time steps. For hyperparameters, we set the number of iterations for optimization to 20 and the Gumbel-Softmax temperature to 1.0. The forget coefficient is set to 0.7. We adopt the dynamic learning rate scheduling described in the Appendix E.2. The base Adam learning rate $l_{\text{Adam, base}}$ is set to 0.3, and the KL divergence weight $\eta_{\text{KL}}$ is set to 0.05. Additionally, we set $\lambda_{\text{KL, schedule}}$ and $\lambda_{\text{lr, schedule}}$ to 0.0 and 1.0, respectively.

---

[14]https://github.com/exitudio/MMM

Measurement  DPS  DDRM  PSLD  ReSample  **G2D2 (Proposed)**  Ground truth

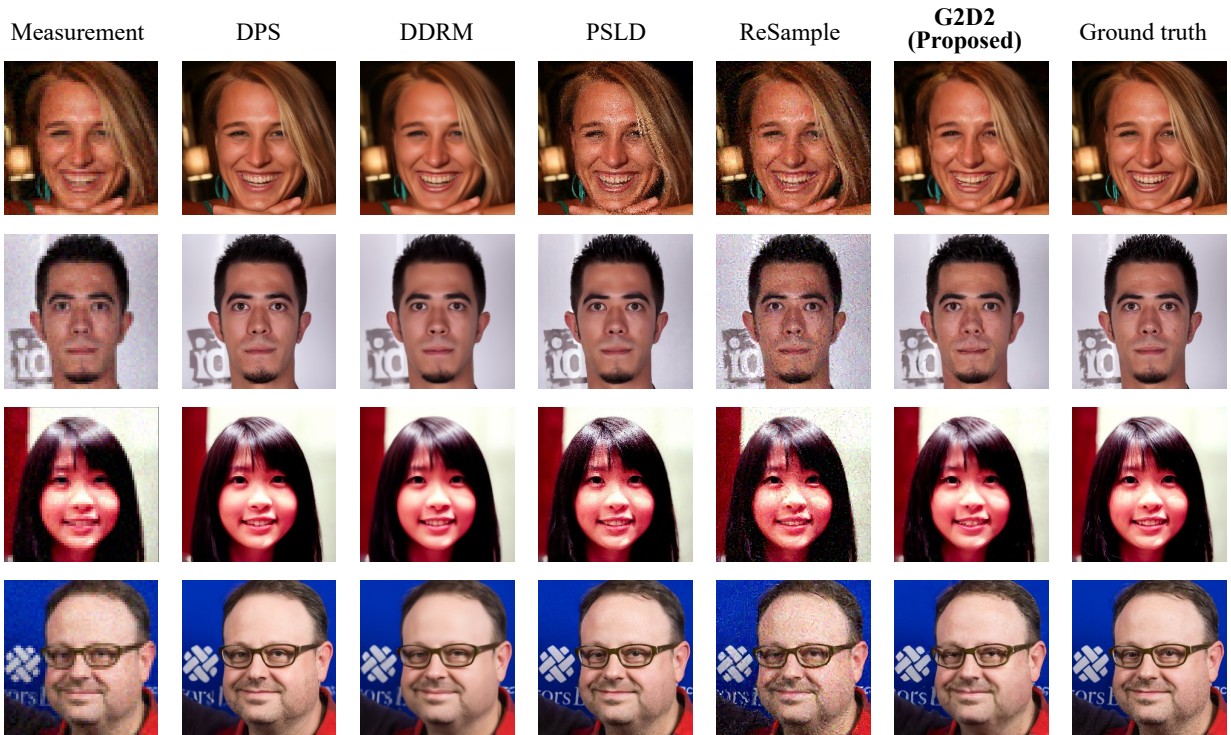

Super resolution (x4, $\sigma = 0.05$)

Figure 12: Qualitative results of G2D2 and comparison methods.

Measurement    DPS    DDRM    PSLD    ReSample    **G2D2 (Proposed)**    Ground truth

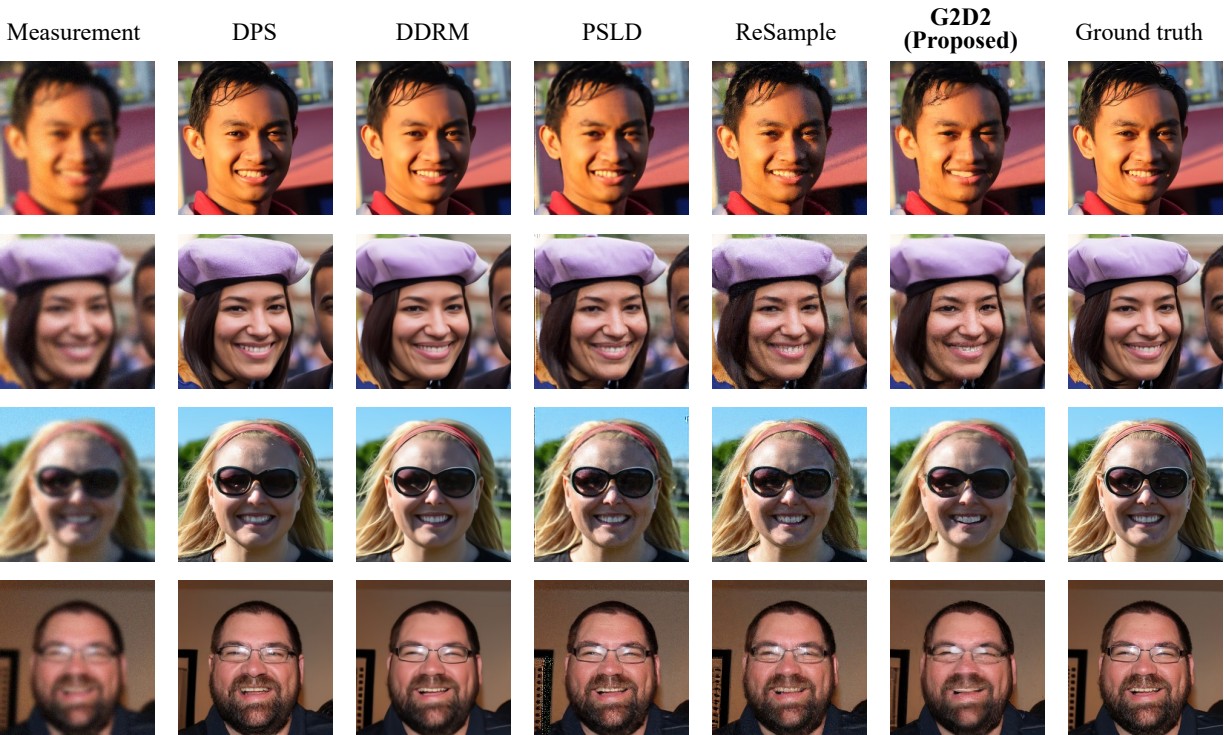

Gaussian Deblur ($\sigma = 0.05$)

Figure 13: Qualitative results of G2D2 and comparison methods.

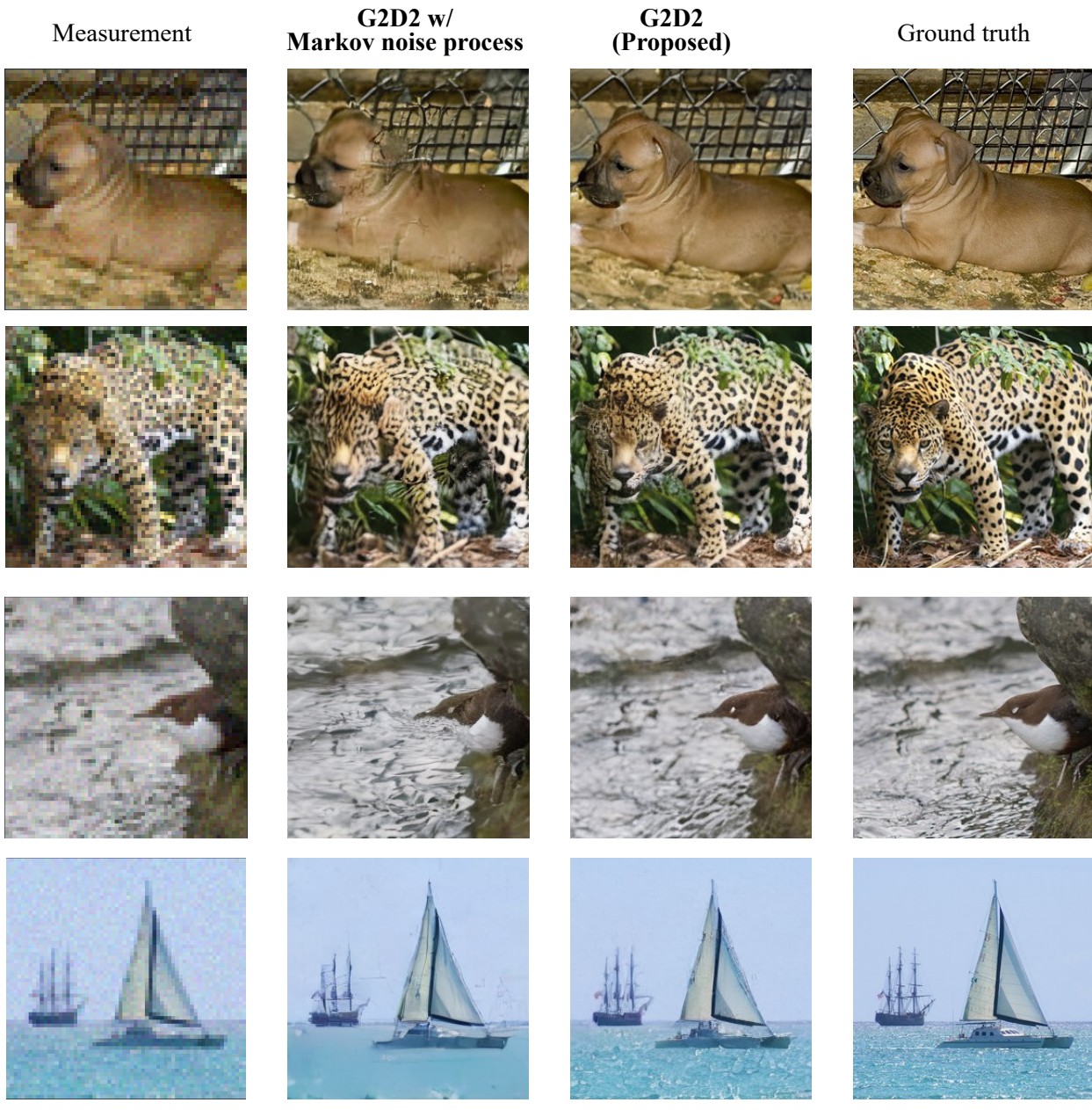

Super resolution (x4, $\sigma = 0.05$)

Figure 14: Qualitative results comparing G2D2 and G2D2 with Markov noise process (Super-resolution task).

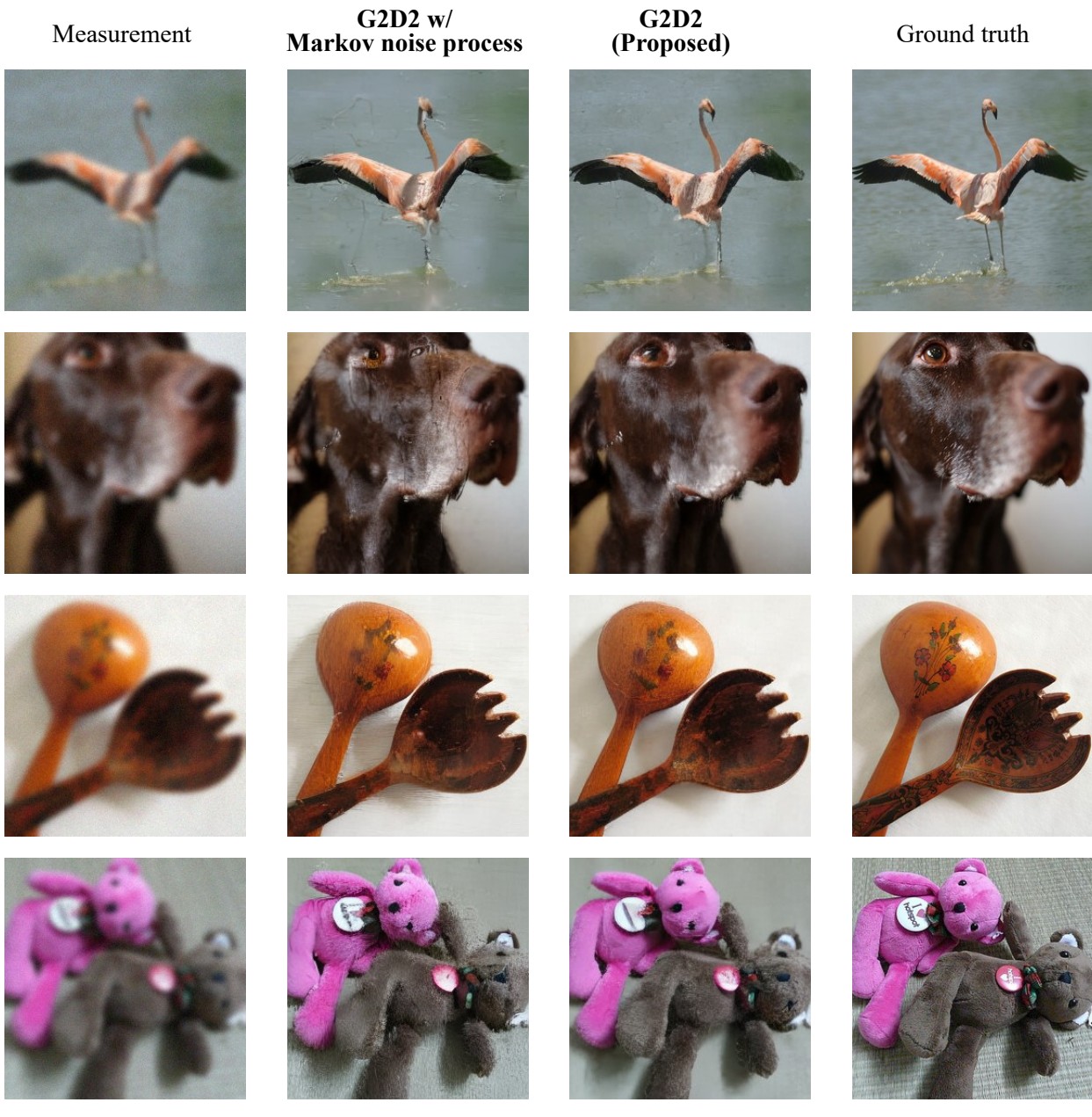

Gaussian Deblur ($\sigma = 0.05$)

Figure 15: Qualitative results comparing G2D2 and G2D2 with Markov noise process (Gaussian Deblurring task).

