# OpenReview forum: "G2D2: Gradient-Guided Discrete Diffusion for Inverse Problem Solving"
_TMLR — Accepted by TMLR_

### Review · Reviewer_ZB2Y · 2025-05-03

**Summary Of Contributions:**

The authors provide a novel posterior sampling algorithm, Gradient‑Guided Discrete Diffusion (G2D2), for solving linear inverse problems using a discrete diffusion prior. This problem is challenging because of the non‑differentiable nature of the model. The authors overcome these technical difficulties by introducing continuous relaxation and optimizing a variational distribution. The authors prove this variational distribution is bounded. In addition, to handle the masking scheme introduced in the architecture of many discrete models, they introduce a star‑shaped noise process, which significantly improves reconstruction when solving the proposed algorithm. The authors provide strong empirical evidence, achieving competitive results against contemporary state‑of‑the‑art baselines while using significantly less GPU memory.

**Audience:**

Yes

**Broader Impact Concerns:**

I do not have any concerns.

**Claims And Evidence:**

Yes

**Requested Changes:**

1. Could the authors consider evaluating their method on a larger set of images to strengthen the comparison with baseline methods?

2. In addition to the point above, could they include a third evaluation metric to provide a more comprehensive assessment?

3. Could the authors clarify the inconsistent results reported in Table 2, as outlined in the strengths and weaknesses section?

**Strengths And Weaknesses:**

### Strengths

1. **Novel impactful prior**
   The paper introduces **G2D2**, to the best of my knowledge the first method that successfully leverages *discrete* diffusion models as priors for inverse problems, broadening the practical scope of diffusion-based techniques.

2. **Strong empirical performance with low memory cost**
   G2D2 is competitive with contemporary state-of-the-art baselines (PSLD, ReSample, DPS, DDRM) while requiring substantially less GPU memory, highlighting both its effectiveness and practicality.

### Weaknesses

* **Limited evaluation set**
Baseline comparisons are performed on only 100 images, which may not be representative of the entire dataset.

* **Narrow metric selection**
  Only two metrics are reported; including additional measures such as **SSIM** or **FID** would give a more rounded view of reconstruction quality.

* **Result inconsistencies**
  There appear to be discrepancies between the reconstruction errors reported in *Table 2* and those in *Tables 5 and 6* (for example, ReSample in the OpenReview paper "[https://openreview.net/pdf?id=j8hdRqOUhN](https://openreview.net/pdf?id=j8hdRqOUhN)").

---

> ### Author Response · Authors · 2025-05-30
> **Rebuttal 1/2**
>
> **The evaluation set is limited to only 100 images, which may not be representative of the entire dataset.
> Only two metrics (PSNR and LPIPS) are reported, whereas additional measures such as SSIM or FID would provide a more comprehensive assessment.**
>
> **[Evaluation expanded to 1k images & more metrics; overall trends remain consistent.]**
> - We appreciate the reviewers' feedback regarding limitations in our evaluation set size and metrics diversity. Our evaluation pipeline partially follows the DAPS paper, and we had conducted evaluations on a partial dataset of 100 images because many comparison methods require several minutes to over 10 minutes per image for estimation.
>
> - However, we agree with the reviewers' comments and, in response, we have expanded our evaluation to 1,000 images each from the ImageNet and FFHQ datasets. We have also introduced Structural Similarity Index Measure (**SSIM**), Fréchet Inception Distance (**FID**), and Kernel Inception Distance (**KID**) in addition to the previously reported Peak Signal-to-Noise Ratio (PSNR) and Learned Perceptual Image Patch Similarity (LPIPS). First, we would like to note that even with the expanded dataset and additional metrics, the overall trends do not significantly change and remain consistent with our original results.
>
> - In particular, evaluating certain baselines such as PSLD takes approximately 10 minutes per image. This means that evaluating a single task and dataset consisting of 1,000 images requires about one week on a single A6000 GPU. Therefore, to provide timely feedback, we first report additional results for G2D2 and key baselines on the 1,000-image ImageNet set. The evaluation of the remaining methods on FFHQ is ongoing and will be integrated into the manuscript upon completion.
>
> **[Results on 1k ImageNet PSNR/LPIPS/SSIM: matching earlier findings]**
>
> **Super Resolution**
>
> | Prior Type | Method | LPIPS↓ | PSNR↑ | SSIM↑ | FID↓ | KID↓ |
> | --- | --- | --- | --- | --- | --- | --- |
> | Pixel-domain | DPS | 0.362 | 22.67 | 0.618 | 46.0 | 4.14e-03 |
> |  | DDRM | 0.351 | 24.27 | 0.669 | 52.3 | 1.16e-02 |
> | LDM | PSLD | 0.331 | 24.02 | 0.675 | 47.9 | 8.26e-03 |
> |  | ReSample | 0.373 | 23.31 | 0.616 | 68.8 | 1.96e-02 |
> | Discrete | G2D2 (proposed) | 0.340 | 23.82 | 0.638 | 50.0 | 8.65e-03 |
>
> **Gaussian Deblurring**
>
> | Prior Type | Method | LPIPS↓ | PSNR↑ | SSIM↑ | FID↓ | KID↓ |
> | --- | --- | --- | --- | --- | --- | --- |
> | Pixel-domain | DPS | 0.432 | 19.49 | 0.473 | 62.0 | 8.58e-03 |
> |  | DDRM | 0.250 | 27.61 | 0.786 | 34.0 | 4.04e-03 |
> | LDM | PSLD | 0.359 | 23.95 | 0.659 | 51.0 | 9.56e-03 |
> |  | ReSample | 0.425 | 23.07 | 0.586 | 82.6 | 2.44e-02 |
> | Discrete | G2D2 (proposed) | 0.367 | 23.37 | 0.604 | 61.6 | 1.39e-02 |
>
> - From the evaluation on the 1,000-image ImageNet subset, the following trends are observed. For PSNR and LPIPS, G2D2 performs slightly worse than PSLD but outperforms DPS in many cases, consistent with our previous 100-image experiments (Tables 1 and 2). The newly added SSIM metric generally maintains the same relative ordering.
>
> - For FID and KID, which capture perceptual quality and distribution similarity, we observe a reversal in performance rankings compared to metrics like LPIPS. This is particularly evident when comparing DPS to PSLD (which uses continuous latent diffusion models as a prior) and G2D2 (which uses discrete diffusion models as a prior). This reversal might be attributed to the fact that the pre-trained model used by DPS was specifically trained on the ImageNet dataset, while the pre-trained models used by PSLD and G2D2 were trained on more diverse image datasets (e.g., ITHQ dataset).
>
> - **These results confirm that even with larger datasets and a broader set of metrics, G2D2 maintains its practical advantage of requiring significantly less GPU memory while providing performance comparable to existing methods.**

---

> > ### Author Response · Authors · 2025-05-30
> > **Rebuttal 2/2**
> >
> > **There are discrepancies between reconstruction errors reported in Table 2 and those in Tables 5 and 6 of the ReSample paper.**
> >
> > **[Numerical differences with ReSample paper due to G2D2's different task setup (higher noise level: 0.05 vs. ReSample's 0.01)]**
> > - As the reviewer pointed out, there are differences in the numerical values between the two papers. This is because we evaluated all methods under more challenging conditions with higher noise levels. Specifically, in the G2D2 paper, we consistently evaluated all comparison methods with a noise standard deviation of 0.05. In contrast, the corresponding Table 6 of the ReSample paper used a noise standard deviation of 0.01. Since reconstruction becomes more difficult with higher noise levels, ReSample's performance values appear lower in our G2D2 paper than those reported in the original ReSample paper.

---

> > > ### Author Response · Authors · 2025-06-24
> > > **Added the FFHQ results and revised the manuscript.**
> > >
> > > In addition to the evaluation on 1000 images from ImageNet, we report additional evaluations on 1000 images from the FFHQ validation set. The trends for existing metrics (regarding PSNR and LPIPS) remain unchanged compared to when we used 100 images. The SSIM results also maintain the expected order. However, the FID and KID scores appear to be inferior to PSLD. This is likely dependent on the performance of the pre-trained model.

---

> > > > ### Author Response · Authors · 2025-06-24
> > > >
> > > > **[Results on 1k FFHQ]**
> > > >
> > > > **Super Resolution**
> > > >
> > > > | Prior Type | Method | LPIPS↓ | PSNR↑ | SSIM↑ | FID↓ | KID↓ |
> > > > |------------|--------|--------|-------|-------|------|------|
> > > > | Pixel-domain | DPS | 0.238 | 26.07 | 0.756 | 26.85 | 0.67e-02 |
> > > > | | DDRM | 0.252 | 28.09 | 0.804 | 46.19 | 3.00e-02 |
> > > > | LDM | PSLD | 0.282 | 27.12 | 0.757 | 37.37 | 2.07e-02 |
> > > > | | ReSample | 0.508 | 23.07 | 0.445 | 85.86 | 7.05e-02 |
> > > > | Discrete | G2D2 (proposed) | 0.265 | 27.29 | 0.763 | 45.21 | 3.08e-02 |
> > > >
> > > > **Gaussian Deblurring**
> > > >
> > > > | Prior Type | Method | LPIPS↓ | PSNR↑ | SSIM↑ | FID↓ | KID↓ |
> > > > |------------|--------|--------|-------|-------|------|------|
> > > > | Pixel-domain | DPS | 0.234 | 25.47 | 0.731 | 24.10 | 0.40e-02 |
> > > > | | DDRM | 0.209 | 30.89 | 0.863 | 42.51 | 2.51e-02 |
> > > > | LDM | PSLD | 0.307 | 26.96 | 0.750 | 38.84 | 1.99e-02 |
> > > > | | ReSample | 0.336 | 25.91 | 0.642 | 45.26 | 2.85e-02 |
> > > > | Discrete | G2D2 (proposed) | 0.280 | 26.91 | 0.743 | 47.70 | 3.46e-02 |

---

### Review · Reviewer_acGE · 2025-05-07

**Summary Of Contributions:**

This paper presents a new algorithm to solve inverse problems in discrete space using a discrete diffusion model. The proposed algorithm implements a "star-shaped" diffusion sampling process, which recursively samples $z_t \sim p(z_t|y)$, $z_0 \sim p(z_0|z_t,y)$ with annealing noise levels. Samples from $p(z_0|z_t, y)$ are obtained by variational inference, where the log-likelihood $\mathbb E_{z_0\sim p_\alpha} \log p(y|z_0)$ is efficiently estimated using the Gumbel-softmax reparameterization trick. The proposed algorithm is validated using VQ-diffusion models trained on VQ latent space of images, yielding comparable results on image restoration tasks as previous works that exploit continuous diffusion models.

**Audience:**

Yes

**Broader Impact Concerns:**

None.

**Claims And Evidence:**

Yes

**Requested Changes:**

The manuscript looks great in its current version, but I believe addressing the questions above in more detail could further enhance its clarity and impact. For example, it would be good to elaborate more on whether G2D2 could be applied to categorical distributions without an underlying continuous representation.

**Strengths And Weaknesses:**

### Strength:
This work studies an interesting topic of diffusion posterior sampling with discrete diffusion models, which to my knowledge is less explored in the literature. The interpretation of the star-shaped diffusion process provides more theoretical clarity on top of previous works (Okhotin et al. 24, Zhang et al. 24). The experimental results is also interesting, which shows the potential of discrete diffusion models in solving inverse problems that lie in continuous space.

### Drawback:
A potential drawback of the proposed method might be its reliance on the Gumbel-softmax trick. To make this trick work, the categorical distribution has to have an underlying continuous representation, where the softmax of discrete tokens is meaningful. This might potentially limit the scope of the proposed algorithm.

### Some questions:
- As claimed in the paper, G2D2 is targeted at linear inverse problems. However, it seems that the only requirement of the forward model is to query its log likelihood as shown in Equation (8). Is there a particular reason for restricting G2D2 to linear inverse problems?

- The ablation study on the graphical model seems interesting to me. Could the author provide more details on implementing G2D2 with Markov noise process? Is G2D2 w/ Markov noise an analogy of DPS for discrete diffusion models while G2D2 is more analogous to DAPS?

---

> ### Author Response · Authors · 2025-05-30
> **Rebuttal 1/3**
>
> **Dependence on the Gumbel-Softmax may restrict applicability to categorical variables that lack continuous representations.**
>
> **[G2D2 Core Framework Flexibility Beyond Gumbel-Softmax]**
> It should be noted that the use of Gumbel-Softmax (GS) trick is merely a convenient implementation choice, and G2D2's core framework (approximation of the true posterior distribution by variational distribution and adoption of star-shaped noise processes) is not fundamentally limited to this specific reparameterization trick. Therefore, it is theoretically possible to replace it with gradient estimators for discrete variables such as REINFORCE or Straight-Through Estimator, which we position as future extension challenges.
>
> **[Justification for Current Gumbel-Softmax Use in Continuous Domains]**
> In our current implementation, as the reviewer pointed out, we use a continuous relaxation based on the GS trick, which assumes a "meaningful continuous relaxation" through codebook embeddings. Indeed, for the data domains primarily addressed in inverse problems (images, motion, audio), this type of continuous relaxation is commonly used, and such an assumption does not significantly narrow our scope.
>
> **[Consideration of Inherently Discrete Domains and Scope Limitation]**
> Furthermore, regarding inherently discrete domain data such as text or protein sequences, their generative models may also internally utilize relaxed continuous representations. For instance, previous research by (Gruver, et al., 2024) attempts to control the generation of data with specific features by optimizing these continuous representations. While such approaches are interesting, they cannot necessarily be mapped to the inverse problem formulations we address, and thus fall outside the scope of this paper.
>
> (Gruver, et al., 2024) "Protein design with guided discrete diffusion", NeurIPS2024.

---

> > ### Author Response · Authors · 2025-05-30
> > **Rebuttal 2/3**
> >
> > **The manuscript should explain why the method is confined to linear inverse problems even though only the forward-model log-likelihood is required.**
> >
> > **[Rationale for Focusing on Linear Inverse Problems and Potential for Nonlinear Extension]**
> > In this paper, we focused primarily on linear inverse problems to verify the effectiveness of G2D2. This was done to clearly demonstrate the basic performance of the method and to facilitate comparisons with standard benchmarks. However, as the reviewer correctly pointed out, the likelihood term $-\mathbb{E}\_{\mathbf{z}\_{0}\sim \tilde{p}\_{\alpha}(\mathbf{z}\_{0}|\mathbf{z}\_{t}, \mathbf{y})}\left[\log q\_{\text{star}}(\mathbf{y}|\mathbf{z}\_{0})\right]$ in equation (8) itself does not necessarily require the assumption that the forward model is linear. In principle, the extension to nonlinear problems is possible as long as the likelihood function and its gradient are computable. As mentioned in the conclusion section, we would like to leave this as future work.

---

> > > ### Author Response · Authors · 2025-05-30
> > > **Rebuttal 3/3**
> > >
> > > **Implementation details of the Markov-noise version of G2D2 and its correspondence to DPS/DAPS should be described clearly.**
> > >
> > > **[Implementation Details of G2D2 with Markov Noise Process and Its Limitations]**
> > > We agree with and appreciate the reviewer's comments. To supplement the implementation details regarding the G2D2 variant using the Markov noise process discussed in Sec. 4.3 of the paper, we have expanded the explanation in Appendix E.3 (and Algorithm 2 within that section) in the updated manuscript. This variant uses the posterior distribution of the conventional Markovian noise process instead of the star-shaped noise process (corresponding to line 10 in Algorithm 2). This approach means that "re-masking" operations are not allowed in the reverse process, making it difficult to correct errors in the early stages.
> > >
> > > **[Analogies to DPS and DAPS for Markov and Star-Shaped G2D2 Variants]**
> > > We find the reviewer's observation about similarities to DPS/DAPS very insightful. Conceptually, G2D2 with Markov noise process can be viewed as an approach that provides guidance along the original model's generation process, similar to DPS. On the other hand, as the reviewer pointed out, G2D2 with the star-shaped noise process has similarities to DAPS (Zhang et al., 2024). This approach enables more flexible error correction by adopting a graphical model with a star-shaped noise process at sampling time for inverse problems, different from regular sampling for generation.
> > >
> > > **[Significance of Star-Shaped Process for Discrete Diffusion Overcoming Mask-Absorbing Constraints]**
> > > Nevertheless, we would like to emphasize that the contribution of the star-shaped noise process is more significant in the case of G2D2. In DPS, which is formulated in the continuous domain, error correction is still possible through gradient updates that can modify the entire image, although it might be more difficult to correct initial errors compared to DAPS. In contrast, this is impossible with mask-absorbing discrete diffusion models, as tokens once determined cannot return to the [MASK] state. The uniqueness of our method lies in introducing these concepts in the context of discrete diffusion models and effectively utilizing the star-shaped noise process to overcome the constraints of models with mask-absorbing states.
> > >
> > > (Zhang et al., 2024) "Improving diffusion inverse problem solving with decoupled noise annealing", CVPR2025.

---

### Review · Reviewer_U988 · 2025-05-15

**Summary Of Contributions:**

This paper addresses the problems of  discrete diffusion models with discrete latent codes in their application to inverse problems formulated in continuous spaces.  It firstly addresses the problem of non-differentiability in discrete diffusion models by proposing Gradient-Guided Discrete Diffusion (G2D2), an inverse problem solving method that uses a discrete diffusion model as a prior,  focused on discrete latent variables such as those found in vector-quantized (VQ)-VAE models. It then addresses the  limitation of mask-absorbing processes of discrete diffusion models by incorporating the star-shaped noise process previously proposed in the context of continuous diffusion models.  Experiments conducted on the several standard datasets, demonstrate preliminary  that G2D2 achieves comparable performance to continuous counterparts.

**Audience:**

Yes

**Claims And Evidence:**

Yes

**Requested Changes:**

(1) provide the wall-clock times to compare the efficiency in the experiments.

(2) provide the dimensions  of  $Q_t$ and vector $\mathbf{v}(z_{t,i})$  in the descriptions

**Strengths And Weaknesses:**

**Strength:**



This paper is overall well written,  and the motivation is clear. This paper provides many detail to illustrate the problems of  Discrete diffusion models and how the proposed methods address the problems.  The idea to use star-shaped noise process for Discrete diffusion models make sense, and this paper provides derivation and theorem to support it.



**Weaknesses:**



The main weaknesses of this paper are the empirical results.

(1) The proposed Star-shaped noise process intuitively will add the time-steps for generating an/a image/sample, therefore add the wall-clock times.  Even though this paper mentions that "maintaining competitive computational speed among gradient-based methods", it does not quantitative results. This paper should provide the wall-clock times to compare the efficiency in the experiments.

(2) The experiments are based on  "a  pretrained VQ-Diffusion model", how about the results on other models? Besides, is it possible to train the model from scratch for comparing other methods?



Other minors:

(1) It is not clear what is the shape of matrix $Q_t$ and vector $\mathbf{v}(z_{t,i})$ in Eqn.1 and 2 , it is better to provide the dimensions  of  $Q_t$ and vector $\mathbf{v}(z_{t,i})$  in the descriptions

---

> ### Author Response · Authors · 2025-05-30
> **Rebuttal 1/**
>
> **Quantitative results (wall-clock time) should be provided to demonstrate the computational efficiency impact of the Star-shaped noise process.**
>
> We thank the reviewer for the feedback.
>
> **[Star-shaped noise process doesn't increase wall-clock time and complexity]**
> - First, we would like to clarify that the introduction of the Star-shaped noise process has not led to an increase in wall-clock time compared to its Markovian counterpart. This is because their difference lies only in the computation of the distribution of $\mathbf{z}\_{t-1}$ given the categorical distribution of $\mathbf{z}\_{0}$. We will now explain why the Star-shaped noise process does not increase computational complexity. Comparing Algorithm 1 (G2D2 algorithm) in the main text with Algorithm 2 (G2D2 with Markov Noise Process) in Appendix E.3, the difference between these two is in line 10, where $\mathbf{z}\_{t-1}$ is sampled using the optimized $\tilde{p}\_{\alpha}$. At this point, G2D2 performs sampling using $q\_{\text{star}}(\mathbf{z}\_{t-1}|\mathbf{z}\_{0})$, which does not use information from the previous step $\mathbf{z}\_{t}$, whereas Algorithm 2 performs sampling using the posterior distribution $q\_{\text{Markov}}(\mathbf{z}\_{t-1}|\mathbf{z}\_{0}, \mathbf{z}\_{t})$. As seen in various implementations including VQ-Diffusion, the difference in computational complexity for this part is minimal.
>
> **[Computational Efficiency and Wall Clock Time]**
> - Regarding the wall clock time, we have included information about G2D2's computational efficiency in Appendix E.5, Table 5. This table compares GPU memory usage and wall-clock time between G2D2 and other baseline methods (DPS, DDRM, PSLD, ReSample) for the Gaussian deblurring task on ImageNet using a single NVIDIA A6000 GPU. G2D2 demonstrates that it maintains competitive computational speed among gradient-based methods while consuming the least amount of GPU memory.
>
> | Method | GPU Memory Usage (GiB) | Wall-Clock time (s) |
> |--------|------------------------|---------------------|
> | G2D2 (Proposed) | 4.7 | 194 |
> | DPS | 10.7 | 277 |
> | DDRM | 5.8 | 4 |
> | PSLD | 20.9 | 738 |
> | ReSample | 7.1 | 555 |
>
> - In the revised manuscript, we have included this discussion to further clarify the efficiency of our method (see Appendix. E.3).

---

> > ### Author Response · Authors · 2025-05-30
> > **Rebuttal 2/n**
> >
> > **Experiments are based on a single pre-trained VQ-Diffusion model, requiring results on other models or comparisons with models trained from scratch.**
> >
> > We appreciate your valuable suggestions.
> >
> > **[Training-Free Methodology]**
> > - First, we note that comparisons between training from scratch and other methods are not included in this study, as they deviate from the main purpose of our research, which focuses on utilizing pre-trained models. While both approaches have their merits and demerits, the advantage of training-free methods is that they do not require additional training for each task.
> >
> > - Next, we clarify that our method is training-free, following a common setup in diffusion-based inverse problem solvers. The goal is to sample from $p(x | y)$ given an observation $y$. By Bayes' rule, $p(x | y) \propto p(x)p(y | x)$, allowing us to use a pre-trained model as a proxy for the prior $p(x)$. The algorithm design thus focuses on aligning with the observation model $p(y | x_0)$.
> >
> > **[Verification across Different Pre-trained Priors]**
> > - While the choice of pre-trained models (e.g., motion models) may influence the prior distribution, this is not part of the algorithmic design and therefore should not be considered in the evaluation of the algorithm itself. In our paper, in addition to experiments using VQ-Diffusion as a prior distribution for images (Sec. 4.2), we also conducted experiments with models for motion generation (Sec. 4.4), verifying our approach with multiple pre-trained models.
> >
> > **[Additional Experiments: G2D2 with CLIP-VQDiffusion Prior]**
> > - In order to further examine the effectiveness of our methods across various pre-trained models for the same task, we have added experiments evaluating G2D2 with an alternative off-the-shelf pre-trained model, CLIP-VQDiffusion (Han & Kim, 2024), for the same image inverse problem tasks (super-resolution and Gaussian deblurring on the FFHQ dataset). A detailed comparison with the VQ-Diffusion prior trained on ITHQ dataset, including quantitative results (Table 7) and qualitative examples (Figure 8), is now provided in the updated Appendix E.8.
> >
> > **[Analysis of CLIP-VQDiffusion Results]**
> > - Interestingly, our results show that VQ-Diffusion trained on ITHQ (hereafter VQDiff-ITHQ) achieved slightly better quantitative performance on FFHQ tasks compared to CLIP-VQDiffusion trained on FFHQ (hereafter CVQDiff-FFHQ). Qualitatively, while results using CVQDiff-FFHQ as a prior tended to produce fewer artifacts, its reconstructions showed a tendency to deviate slightly from the ground truth images compared to those using VQDiff-ITHQ.
> >
> >
> > Table 7 from the revised manuscript
> > | Dataset | Prior Model | SR (×4) |  | Gaussian Deblurring |  |
> > |---------|-------------|---------|---------|---------------------|---------|
> > |         |             | PSNR↑   | LPIPS↓  | PSNR↑               | LPIPS↓  |
> > | FFHQ    | **VQDiff-ITHQ** (Tang et al., 2022) | **27.44** | **0.259** | **27.00** | **0.273** |
> > |         | **CVQDiff-FFHQ** (Han et al., 2024) | 25.32 | 0.283 | 24.41 | 0.289 |
> >
> > - Since CVQDiff-FFHQ was trained using FFHQ, it likely has an advantage in prior modeling (corresponding to the first term of the optimization target in Algorithm 1, L10). However, the optimization of the likelihood in the measurement equation (the second term) involves decoder calculations, and since both models use different decoders, this difference may be influencing the results.
> >
> > (Han & Kim, 2024) "CLIP-VQDiffusion : Language Free Training of Text To Image generation using CLIP and vector quantized diffusion model", arXiv, 2024.

---

> > > ### Author Response · Authors · 2025-05-30
> > > **Rebuttal 3/3**
> > >
> > > **The dimensions of matrix Qt and vector v(zt,i) in Equations 1 and 2 should be explicitly specified.**
> > >
> > > As the reviewer pointed out, explicit descriptions were lacking. We have added these descriptions in the revised manuscript.
> > >
> > > - The vector $v(z\_{t,i})\in \\{0, 1\\}^{K+1}$ is a one-hot encoded vector representing the $i$-th token at time $t$. It combines $K$ states from VQ-VAE and a special mask state $[\text{MASK}]$.
> > >
> > > - The transition matrix $Q\_{t}\in\mathbb{R}^{(K+1)\times (K+1)}$ determines the transition probabilities between tokens.

---

### Author Response · Authors · 2025-05-30
**General comments to the reviewers**

**First, we would like to express our sincere gratitude to the three reviewers who provided constructive feedback.**

The reviewers acknowledged the **novelty** of this research in utilizing discrete diffusion models for inverse problem solving, as well as its **practical significance**, **GPU memory efficiency**, and **clarity of theoretical organization**. On the other hand, we also received specific suggestions for further improvement, including:
- Quantitative comparison of computation time
- Expansion of evaluation metrics
- Description of symbols and dimensions
- Supplementary explanations regarding the versatility of the method

Based on these comments, we have added additional experiments and detailed discussions to both the main text and appendix, aiming to clarify expressions and improve reproducibility. **Below, we will report our responses to each comment in order.**

We hope that these responses address the reviewers' questions and concerns, but we remain open to further discussion.

---

> ### Author Response · Authors · 2025-06-24
>
> Thank the reviewers for engaging with our paper and providing constructive feedback. Based on the comments from Reviewer ZB2Y, we have expanded the number of evaluation images from **100 to 1000** and uploaded the revised manuscript. Additionally, we have introduced **additional metrics** including SSIM, FID, and KID. Regarding the existing metrics (PSNR, LPIPS), **the trends remain consistent** compared to when we used 100 images. The SSIM results also **maintain the expected order**. For FID and KID, we observe that they sometimes show inverse performance ordering compared to LPIPS, which suggests **a dependency on the prior model's performance**. Similar to other methods, **our approach is training-free, and we expect performance improvements through changes to the prior model**.
>
> If there are any additional points that would benefit from further clarification or discussion, we would be happy to address them.

---

### Decision · Action_Editor_rS4q · 2025-08-03

**Recommendation:** Accept as is

**Audience:**

Yes

**Audience Explanation:**

Yes, this work would be of significant interest to multiple segments of TMLR's audience. The paper addresses the intersection of discrete diffusion models and inverse problem solving, appealing to researchers in generative modeling, computational imaging, and optimization-based inference. The introduction of discrete diffusion models as priors for inverse problems represents a novel direction with practical advantages (competitive performance with substantially lower memory requirements), while the star-shaped noise process innovation has broader implications for discrete diffusion architectures.

**Claims And Evidence:**

Yes

**Claims Explanation:**

Yes, the claims are well-supported by accurate, convincing and clear evidence. The authors provide solid theoretical foundations for their approach. The experimental validation is comprehensive, having been expanded from initial 100-image evaluations to 1000 images across multiple datasets with diverse metrics. The key claims about competitive performance and memory efficiency are substantiated through detailed comparisons with state-of-the-art baselines. The computational efficiency claims are backed by wall-clock timing data, and the theoretical advantages of the star-shaped process over mask-absorbing limitations are clearly demonstrated both analytically and empirically.